# The meiotic TERB1-TERB2-MAJIN complex tethers telomeres to the nuclear envelope

Yan Wang[1], Yanyan Chen[1], Juan Chen[2,3], Lijun Wang[1], Leitong Nie[4], Juanjuan Long[1], Haishuang Chang[2,3], Jian Wu[2,3], Chenhui Huang [ID] [2,3] & Ming Lei [ID] [2,3,5]

During meiotic prophase I, telomeres attach to and move on the nuclear envelope (NE), regulating chromosome movement to promote homologous pairing. Meiosis-specific proteins TERB1, TERB2 and MAJIN play a key role in this process. Here, we report the crystal structures of human TERB1-TERB2 and TERB2-MAJIN subcomplexes. Specific disruption of the TERB1-TERB2 or the TERB2-MAJIN interaction in the mouse *Terb2* gene abolishes the telomere attachment to the NE and causes aberrant homologous pairing and disordered synapsis. In addition, depletion of SUN1 also partially disrupts the telomere-NE connection. We propose that the telomere-TRF1-TERB1-TERB2-MAJIN-NE interaction network and the telomere-LINC complex connection are likely two separate but cooperative pathways to stably recruit telomeres to the NE in meiosis prophase I. Our work provides a molecular model of the connection between telomeres and the NE and reveals the correlation between aberrant synapsis and the defective telomere attachment to the NE.

[1] National Center for Protein Science Shanghai, State Key Laboratory of Molecular Biology, CAS Center for Excellence on Molecular Cell Science, Institute of Biochemistry and Cell Biology, Chinese Academy of Sciences, University of Chinese Academy of Sciences, 200031 Shanghai, China. [2] Ninth People's Hospital, Shanghai Jiao Tong University School of Medicine, 200011 Shanghai, China. [3] Shanghai Institute of Precision Medicine, 200125 Shanghai, China. [4] National Key Laboratory of Crop Genetic Improvement, Huazhong Agriculture University, 430072 Wuhan, China. [5] Key laboratory of Cell Differentiation and Apoptosis of Chinese Ministry of Education, Shanghai Jiao Tong University School of Medicine, 200025 Shanghai, China. These authors contributed equally: Yan Wang, Yanyan Chen. Correspondence and requests for materials should be addressed to C.H. (email: huangchh@shsmu.edu.cn) or to M.L. (email: leim@shsmu.edu.cn)

Meiosis produces reproductive cells by two consecutive rounds of nuclear division following a single round of DNA replication[1]. The maintenance of genomic integrity during meiosis depends on the accurate chromosome segregation[2]. Prophase I is the longest and most complex phase in meiosis and is characterized by homologous pairing and formation of synaptonemal complex (SC)[3]. Homologous pairing is essential for both genetic recombination and, more crucially, precise segregation of the homologs in the subsequent cell division step[3,4].

Vertebrate telomeres consist of an array of TTAGGG bound by the evolutionarily conserved shelterin complex[5,6]. SUN1 (Sad1 and UNC84 domain containing 1) and KASH5 (Klarsicht/ANC-1/Syne/homology 5) assemble into the mammalian meiosis-specific LINC (linker of nucleoskeleton and cytoskeleton) complex, providing the binding sites for telomeres on the inner surface of the nuclear envelope (NE)[7–9]. In meiosis prophase I, cells establish the association between telomeres and the LINC complex, bridging chromosomes to the cytoskeleton for force transduction to allow chromosome movements[10,11]. In most organisms, telomeres attach to and move along the NE during leptotene stage and become clustered at the side of the nucleus adjacent to the centrosome at zygotene stage, resulting in a transient bouquet-like arrangement of chromosomes[12,13]. The meiotic telomere dynamics drives the rapid chromosome movements, which is suggested to contribute to homologous searching and resolve unwanted entanglements between non-homologs[10,14].

Another protein complex consisting of TERB1 (telomere repeat binding bouquet formation protein 1), TERB2 (telomere repeat binding bouquet formation protein 2), and MAJIN (membrane anchored junction protein) has been identified to establish a second physical linkage for telomere attachment to the NE[15,16]. MAJIN is a putative transmembrane (TM) protein, localized at the inner surface of the NE[16]. TERB1 is a molecular scaffold that simultaneously interacts with TERB2 and shelterin subunit TRF1[15,16]. Knockout of any components of the TERB1–TERB2–MAJIN (TTM) complex impairs the telomere–NE attachment and results in infertility in both male and female mice[15,16]. Notably, two additional meiosis-related proteins, Speedy A and cyclin-dependent kinase 2 (CDK2), were recently found to localize at meiotic telomeres involved in telomere–NE attachment[17–19], adding even more complexity to the regulation of meiotic telomeres. Except for the recently reported biochemical and structural dissection of the TRF1–TERB1 interaction, the structural basis for the telomere–NE attachment still remains largely elusive[20,21]. Moreover, the sequential order and physiological roles of the two telomere–NE connections, respectively, established by SUN1 and MAJIN is still poorly understood.

In the present study, we characterize the TERB1–TERB2 and TERB2–MAJIN interactions and determine the crystal structures of the N-terminal domains of TERB2 and MAJIN in complex with TERB2-binding motif of TERB1 and MAJIN-binding motif of TERB2, respectively. We generated knock-in mice with the TERB1-binding-deficient or the MAJIN-binding-deficient mutations in the *Terb2* gene. Examination of the meiosis progression reveals same meiotic defects in both mutant mice, including abolished telomere–NE attachment, aberrant homologous pairing, disordered synapsis, and infertility in both sexes. We propose that disruption of telomere–NE connection impairs the homologous searching, which is the underlying reason for subsequent meiosis defects. Our work provides a molecular model of the connection between telomeres and the NE via the TRF1–TERB1–TERB2–MAJIN interaction pathway and reveals the correlation between aberrant synapsis and the defective telomere attachment to the NE.

## Results

### Characterization of TERB1–TERB2 and TERB2–MAJIN interactions.

Meiotic specific proteins TERB1, TERB2, and MAJIN form a stable complex that plays a critical role in regulating the recruitment of telomeres to the NE[15,16]. Our previous study revealed that TRF1 recognizes a unique motif in TERB1 via its TRF homology (TRFH) domain to regulate the telomere–NE attachment and play a vital role in assisting X–Y chromosome pairing during meiosis[21]. In the TTM complex, TERB2 simultaneously interacts with both TERB1 and MAJIN[16]. To gain further insight into the telomere recruitment function of the TTM complex, we first characterized the interaction between TERB1 and TERB2 using bimolecular fluorescence complementation (BiFC) assay. We found that a short fragment of TERB1 consisting of residues 587–636, which is adjacent to the TRF1-binding motif (TERB1$_{TBM}$, residues 642–656) of TERB1, was necessary and sufficient for binding with the N-terminal domain of TERB2 (residues 1–116) (Fig. 1a, b).

We next set out to map the interaction between TERB2 and MAJIN. MAJIN was identified as an inner nuclear membrane protein containing a putative TM domain at its C-terminus[16]. Consistently, ectopically expressed MAJIN exhibited a clear pattern of nuclear membrane localization in U-2 OS cells, whereas TERB2 when ectopically expressed alone displayed a diffused distribution (Fig. 1c). MAJIN was able to recruit the co-expressed TERB2 to the nuclear membrane, confirming the direct interaction between TERB2 and MAJIN (Fig. 1d)[16]. Various TERB2 fragments were then evaluated for their ability to interact with MAJIN (Fig. 1d). Our data revealed that a short fragment of TERB2 at the C-terminus (residues 168–202) is sufficient for interaction with MAJIN (Fig. 1d). Consistent with previous studies[16], gel filtration chromatography analysis using purified proteins showed that the N-terminal fragment of MAJIN (residues 2–149) mediates the interaction with TERB2 (Fig. 1e). Collectively, these results reveal a linear interaction network within the TTM complex (Fig. 2a)[16].

### Crystal structures of TERB1–TERB2 and TERB2–MAJIN complexes.

To reveal the structural basis for the interactions between TERB1 and TERB2 and between TERB2 and MAJIN, we crystallized the TERB1$_{587-659}$–TERB2$_{2-116}$ and the TERB2$_{119-220}$–MAJIN$_{2-149}$ complexes and determined their structures using single-wavelength anomalous dispersion (SAD) method at resolutions of 3.3 and 2.9 Å, respectively (Supplementary Tables 1 and 2). Well-defined electron density maps allowed unambiguous tracing of polypeptide chains TERB1$_{590-649}$, TERB2$_{4-110}$, TERB2$_{174-209}$, and MAJIN$_{2-109}$ in the complexes (Supplementary Fig. 1a–d), whereas the rest of residues are disordered in the crystals and presumably dispensable for the interactions. Notably, the interaction domains revealed by the crystal structures are consistent with those identified by the mapping analyses (Fig. 1a–d). Hereafter, for simplicity, we will refer to TERB2$_{1-116}$ as TERB2$_{NTD}$ (N-terminal domain), TERB1$_{587-636}$ as TERB1$_{T2BM}$ (TERB2-binding motif), MAJIN$_{1-110}$ as MAJIN$_{NTD}$, and TERB2$_{168-202}$ as TERB2$_{MBM}$ (MAJIN-binding motif), respectively (Fig. 2a).

The TERB1$_{T2BM}$–TERB2$_{NTD}$ complex adopts a compact globular fold, resembling a single folding unit (Fig. 2b). TERB2$_{NTD}$ consists of a three-stranded β sheet (β1–β2–β3) at one end and six α helices that flank the β sheet and scatter at the periphery of the structure (Fig. 2b). TERB1$_{T2BM}$ is a

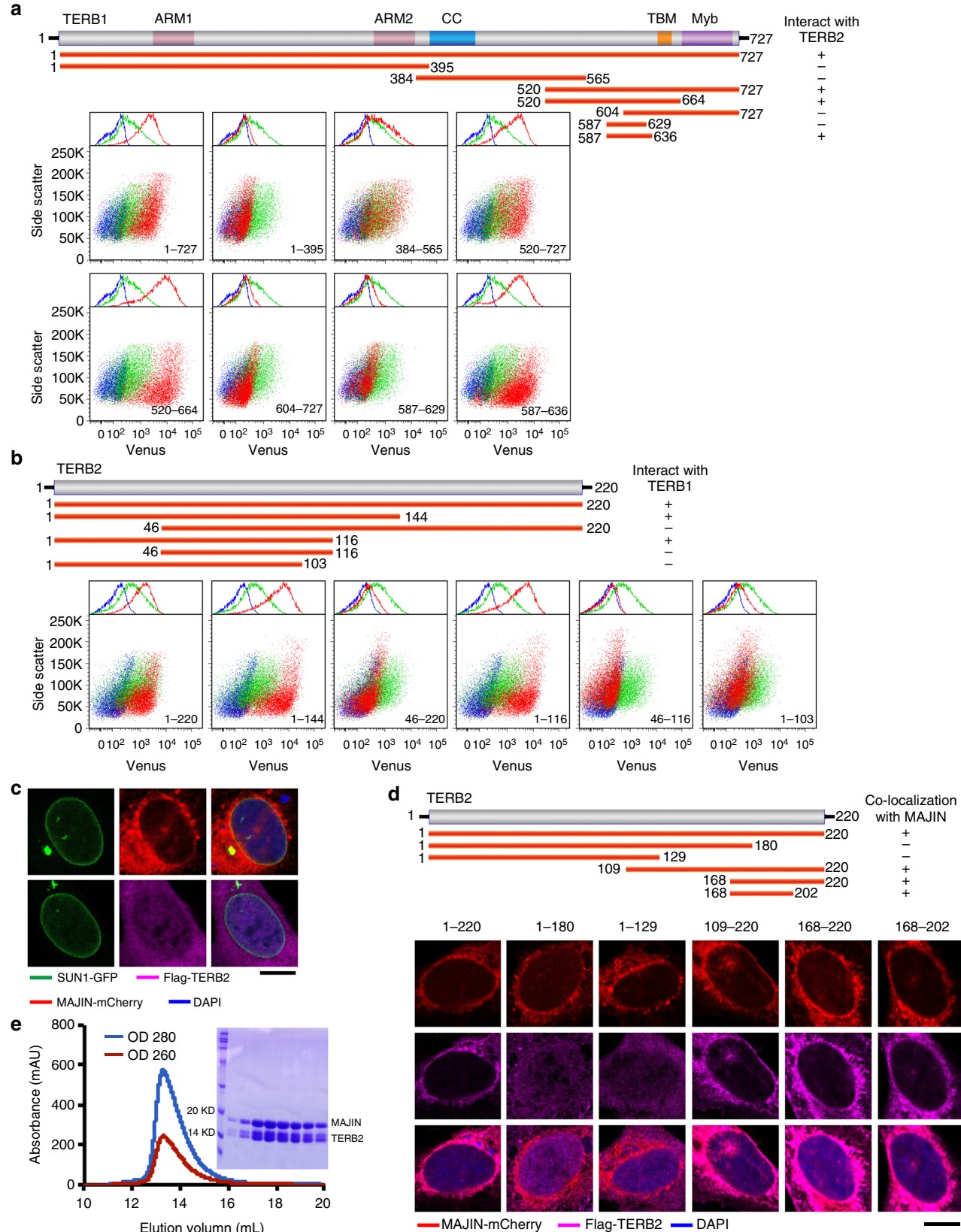

helix–turn–helix (HtH) motif with a long tail that folds back to the middle turn between the two helices (Fig. 2b). Notably, Cys592 and Cys595 at the N-terminus of the tail together with Cys618 and His621 in the middle of the HtH motif chelate a $Zn^2$ $^+$ ion, playing a key role in defining the overall shape of $TERB1_{T2BM}$ (Fig. 2c). $TERB1_{T2BM}$ adopts an extended conformation and packs against the concaved surface of $TERB2_{NTD}$, burying ~1400 $Å^2$ of surface area at the interface (Fig. 2d). The

**Fig. 1** Characterization of the interactions among TERB1, TERB2, and MAJIN. **a** Illustration of different fragments of TERB1 (upper panel). Bimolecular fluorescence complementation (BiFC) data with full-length TERB2 and varies TERB1 fragments (lower panel). Cells expressing TERB2-VC only (blue, as a negative control) or TERB2-VC with -VN vector (green, as a negative control) or TERB2-VC with different TERB1-VN fragments (red) were subjected to fluorescence-activated cell sorting (FACS) analysis. **b** Illustration of various fragments of TERB2 (upper panel). BiFC data with full-length TERB1 and TERB2 fragments (lower panel). Cells expressing TERB1-VC only (blue, as a negative control) or TERB1-VC with -VN vector (green, as a negative control) or TERB1-VC with different TERB2-VN truncations (red) were subjected to FACS analysis. **c** Equator images of U-2 OS cells expressing SUN1-GFP (green) together with mCherry-MAJIN (red) (up) or Flag-TERB2 (magenta) (down). SUN1-GFP was used to indicate the nuclear envelope localization. Cells were stained with 4,6-diamidino-2-phenylindole (DAPI; blue) to show the nuclear position. Scale bars, 10 μm. **d** Schematics of various TERB2 fragments (upper panel). Equator images of U-2 OS cells expressing mCherry-MAJIN (red) and Flag-TERB2 truncations. Cells were stained with Flag antibody (magenta) and DAPI (blue). Scale bars, 10 μm. **e** Gel filtration chromatography profile of the TERB2$_{MBM}$–MAJIN$_{NTD}$ complex. Sodium dodecyl sulfate-polyacrylamide gel electrophoresis image of the peak in the gel filtration chromatography profile is shown

long helix H2 of TERB1$_{T2BM}$ fits into a large hydrophobic groove formed by helices α3 and α6 and loops L$_{33}$ and L$_{56}$ of TERB2$_{NTD}$ through extensive contacts involving a panel of hydrophobic residues at both sides of the interface (Fig. 2c). Helix H1 and the N-terminal tail of TERB1$_{T2BM}$ pack against a flat hydrophobic surface of TERB2$_{NTD}$ formed by the β sheet and helices α3 and α5 (Fig. 2e). In addition to hydrophobic contacts, electrostatic interactions also make substantial contributions to the interactions. Most notably, a cluster of acidic and polar residues in TERB2$_{NTD}$ (Asp55, Glu58, Asp59, and Gln110) mediate a network of salt bridge and hydrogen-bonding interactions with positively charged residues in helices H1 and H2 of TERB1$_{T2BM}$ (Arg605, Arg629, Lys632, and Arg636), helping secure the HtH motif of TERB1$_{T2BM}$ on the surface of TERB2$_{NTD}$ (Fig. 2c, f). Therefore, both hydrophobic and electrostatic interactions are important for the interaction between TERB1$_{T2BM}$ and TERB2$_{NTD}$.

In the TERB2$_{MBM}$-MAJIN$_{NTD}$ complex, MAJIN$_{NTD}$ adopts a compact fold with a curved four-stranded β sheet (β1–β2–β3–β5) holding helix α1 in the center (Fig. 3a). The TERB2$_{MBM}$ polypeptide forms an extended structure that wraps around the surface of MAJIN$_{NTD}$ (Fig. 3a). The driving force for the binding of TERB2$_{MBM}$ to MAJIN$_{NTD}$ is van der Waals interactions (Fig. 3b). The highly hydrophobic TERB2$_{MBM}$ polypeptide meanders along a hydrophobic groove on the surface of MAJIN$_{NTD}$ and buries a total of ~1300 Å$^2$ surface area at the interface (Fig. 3b, c). The C-terminus of TERB2$_{MBM}$ and the loop between strands β3 and β5 of MAJIN$_{NTD}$ contact each other and fold into an intermolecular anti-parallel β sheet that packs onto the central β sheet of MAJIN$_{NTD}$ to further strengthen the interaction (Fig. 3d). It is worthy of noticing that, although there are three TERB2$_{MBM}$–MAJIN$_{NTD}$ complexes in one asymmetric unit in the crystal structure, two protomer complexes apparently form a heterotetramer with an extensive hydrophobic dimeric interface between two MAJIN$_{NTD}$ molecules (Fig. 3e and Supplementary Fig. 1e and 1f). Interestingly, the third protomer form a second heterotetramer with another complex from an adjacent unit cell with the identical hydrophobic interface (Supplementary Fig. 1g). Consistent with a recent study, this observation suggests that TERB2 and MAJIN could form a heterotetramer with a 2:2 stoichiometry[22]. Notably, TRF1 itself forms a stable dimer and binds two TERB1 molecules through its TRFH domain[21]. Therefore, TRF1, TERB1, TERB2, and MAJIN can potentially form a large complex with two sets of each protein with multiple dimeric interfaces (Fig. 3f). Together with previous data[20,21], the crystal structures of the TERB1$_{T2BM}$–TERB2$_{NTD}$ and the TERB2$_{MBM}$–MAJIN$_{NTD}$ complexes provide a molecular model of the telomere–NE connection through the TRF1–TERB1–TERB2–MAJIN interaction network (Fig. 3f).

**Mutational analysis of TERB1–TERB2 and TERB2–MAJIN interfaces.** To corroborate our structural analysis, we examined whether missense mutations of the interface residues of TERB2 could weaken or disrupt the TERB1–TERB2 and TERB2–MAJIN interactions. We examined the interaction of mutant proteins transiently expressed in human embryonic kidney 293T cells. For the TERB1–TERB2 interaction, we focused on a hydrogen bond between Tyr56$^{TERB2}$ and Glu628$^{TERB1}$ in the core of the binary interface (Fig. 4a). A substitution of Tyr56$^{TERB2}$ with a negatively charged glutamate residue would generate an electrostatically unfavorable clash that could severely interfere with the interaction. Consistent with this structural information, co-immunoprecipitation (Co-IP) experiments revealed that, while wild-type (WT) TERB1 and TERB2 showed the expected interaction, the TERB2 Y56E mutation completely abolished the interaction with TERB1 in 293T cells (Fig. 4b). For the TERB2–MAJIN interaction, we focused on the hydrophobic interface between the C-terminal tail of TERB2$_{MBM}$ and the hydrophobic wedge formed by strands S1 in TERB2$_{MBM}$ and β1, β2, and β4 in MAJIN$_{NTD}$, which are critical for stabilization of the interaction (Fig. 4c). In particular, located at the center of this interface, the side chain of Phe192 of TERB2 is buried in a pocket formed by a group of hydrophobic residues of both MAJIN and TERB2 (Fig. 4c). Again consistent with the crystal structure, substitution of Phe192$^{TERB2}$ with a negatively charged bulk arginine residue completely abolished the TERB2–MAJIN interaction in 293T cells (Fig. 4d). These results indicated that single point mutations at the TERB1–TERB2 and TERB2–MAJIN interfaces are sufficient to disrupt the ability of TERB2 to bind TERB1 and MAJIN, respectively. Notably, the Y56E mutation of TERB2 only disrupted the TERB2 interaction with TERB1 but not the one with MAJIN (Fig. 4b, d). And vice versa the F192R mutation only affected the interaction with MAJIN but not with TERB1 (Fig. 4b, d). Therefore, our Co-IP data strongly suggested that these mutations only disrupt the desired interactions of TERB2 in cells. Both Tyr56 and Phe192 are highly conserved residues in TERB2 proteins from different species (Fig. 4e). Consistently, similarly conclusions were obtained when interactions between mouse proteins were examined by Co-IP analysis (Fig. 4f, g).

**Disruption of TERB1–TERB2 and TERB2–MAJIN interfaces in mice.** To elucidate the meiotic role of the TTM complex, we generated knock-in mice with the TERB1-binding-deficient mutation Y56E and the MAJIN-binding-deficient mutation F188R in the *Terb2* gene using the CRISPR–Cas9 method (Supplementary Fig. 2a–d). The heterozygous mice were healthy and fertile and were intercrossed to produce homozygous mice. Heterozygous and homozygous mutant progeny were identified

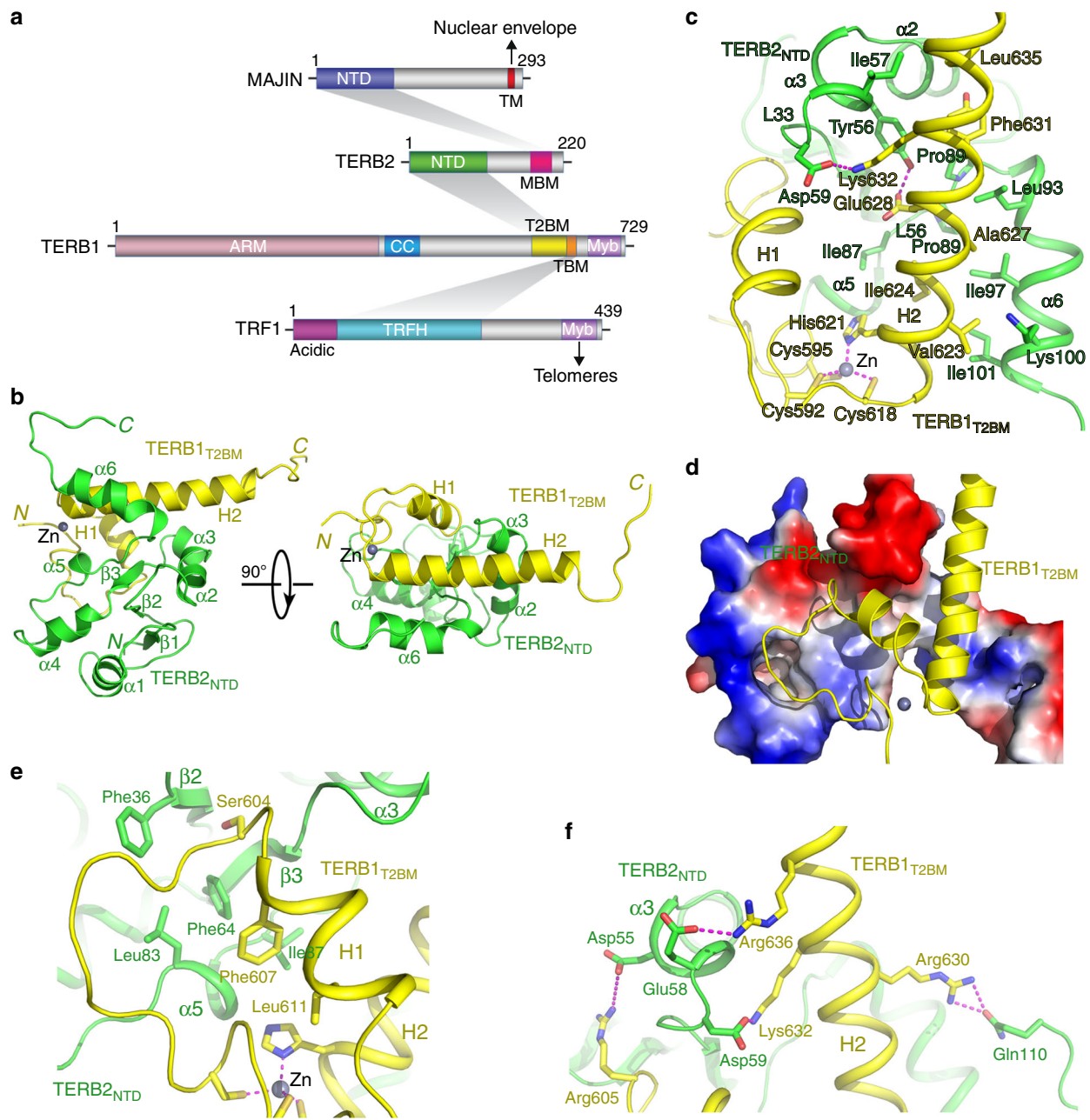

**Fig. 2** Crystal structure of the TERB1$_{T2BM}$–TERB2$_{NTD}$ complex. **a** Domain organization of TRF1, TERB1, TERB2, and MAJIN. Shadings indicate the interactions among these proteins. The interacting domains are labeled and highlighted by different colors. This diagram illustrates the TRF1–TERB1–TERB2–MAJIN interaction network connecting telomeres to the nuclear envelope. Acidic acidic motif, TRFH TRF homology domain, Myb myeloblastosis domain, ARM armadillo repeats, CC coiled coil, TM trans-membrane motif. **b** Ribbon diagrams of two orthogonal views of the TERB1$_{T2BM}$–TERB2$_{NTD}$ complex. TERB1$_{T2BM}$ is colored in yellow and TERB2$_{NTD}$ in green. **c**, **e**, **f** Detailed interactions at the TERB1$_{T2BM}$–TERB2$_{NTD}$ interface. The color scheme is the same as in **b**. Residues important for the interaction are shown in stick models. Salt bridges and hydrogen-bonding interactions are shown as magenta dashed lines. The Zn ion is shown as a gray sphere and residues chelating this Zn ion are shown in stick model in **c**. **d** Electrostatic surface potential of TERB1$_{T2BM}$-binding site of TERB2$_{NTD}$. Positive potential, blue; negative potential, red

by polymerase chain reaction (PCR) and DNA sequencing analyses (Supplementary Fig. 2b and 2d). Pups were born close to the expected Mendelian ratio (Supplementary Table 3), suggesting that the *Terb2* mutations did not impair normal embryonic development. Reverse transcription PCR and DNA sequencing were performed to confirm that mRNA transcripts of the mutant *Terb2* genes carried the desired mutations in homozygous testes (Supplementary Fig. 2b and 2d). Quantitative PCR and western blotting analyses of 2-week-old mouse testes revealed that the

*Terb2* mutations do not alter the in vivo mRNA and protein expression levels of TERB2 (Supplementary Fig. 2e–g). The mRNA levels of TERB1 and MAJIN were also not affected by the *Terb2* mutations (Supplementary Fig. 2e, f).

**Intact TTM complex is required for fertility**. Both male and female mutant mice developed normally into adults with no obvious somatic defects. However, repeated breeding attempts of

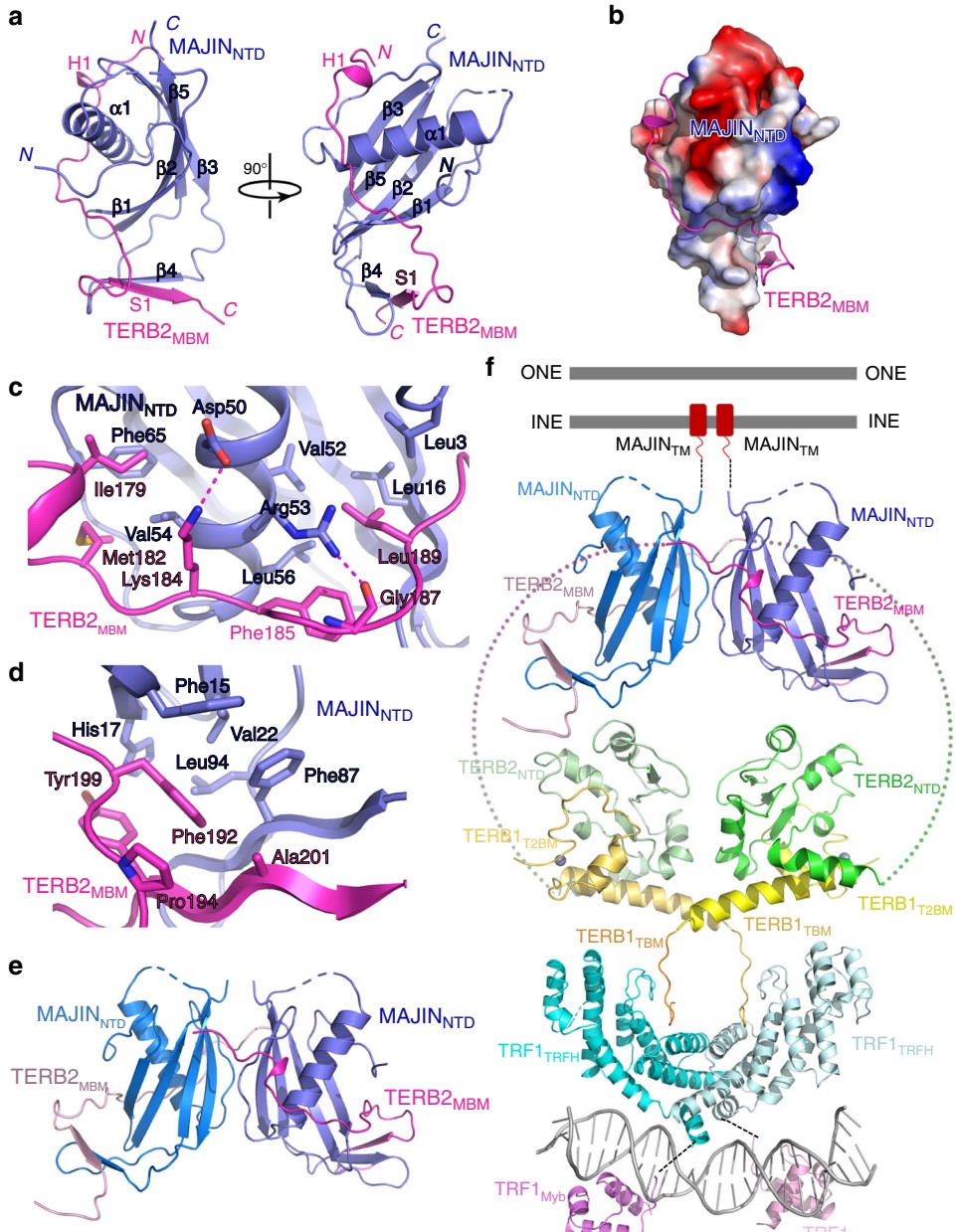

**Fig. 3** Crystal structure of the TERB2_{MBM}–MAJIN_{NTD} complex. **a** Ribbon diagrams of two orthogonal views of the TERB2_{MBM}–MAJIN_{NTD} complex. TERB2_{MBM} is colored in magenta and MAJIN_{NTD} in blue. **b** Electrostatic surface potential of TERB2_{MBM}-binding site of MAJIN_{NTD}. Positive potential, blue; negative potential, red. **c**, **d** Detailed interactions at the TERB2_{MBM}–MAJIN_{NTD} interface. The color scheme is the same as in **a**. Residues important for the interaction are shown in stick models. Salt bridges and hydrogen-bonding interactions are shown as magenta dashed lines. **e** Ribbon diagram shows the potential heterotetramer mode of the TERB2_{MBM}–MAJIN_{NTD} complex revealed in the crystal structure. **f** Structural model of the connection between telomeres and the NE through the TRF1–TERB1–TERB2–MAJIN interaction network

homozygous *Terb2^{Y56E}* and *Terb2^{F188R}* male mice with WT female animals did not yield any offspring (Supplementary Table 4). Although mutant mice exhibited similar body weight compared to their WT littermates, anatomical analysis revealed that the adult testes of homozygous mutant mice were much smaller than those of their WT siblings (Fig. 5a, b). Histological examination showed that adult mutant seminiferous tubules were narrower and devoid of both elongated spermatids and mature sperms (Fig. 5c). Indeed, the epithelium of the majority of the seminiferous tubules was abnormal and contained only a single layer of Sertoli cells and very few spermatogonial cells at the tubule periphery (Fig. 5d). Consequently, there was no mature sperm in either the caput or cauda epididymal lumen of the

homozygous *Terb2^{Y56E}* and *Terb2^{F188R}* mice (Supplementary Fig. 3a). Fluorescence-activated cell sorting (FACS) analysis showed that spermatogenesis process in both *Terb2^{Y56E}* and *Terb2^{F188R}* mice was disrupted at the tetraploid stage of primary spermatocytes (Supplementary Fig. 3b). Furthermore, terminal dexynucleotidyl transferase (TdT)-mediated dUTP nick end labeling (TUNEL) assay revealed obvious positive signals in mutant seminiferous tubules, suggestive of massive apoptotic events that likely account for the absence of post-meiotic germ cells in *Terb2^{Y56E}* and *Terb2^{F188R}* testes (Fig. 5e, f). To further characterize the meiotic defect in *Terb2^{Y56E}* and *Terb2^{F188R}* mice, we examined the meiotic prophase I progression by immuno-fluorescence (IF) staining of the SC proteins SYCP3 and SYCP1.

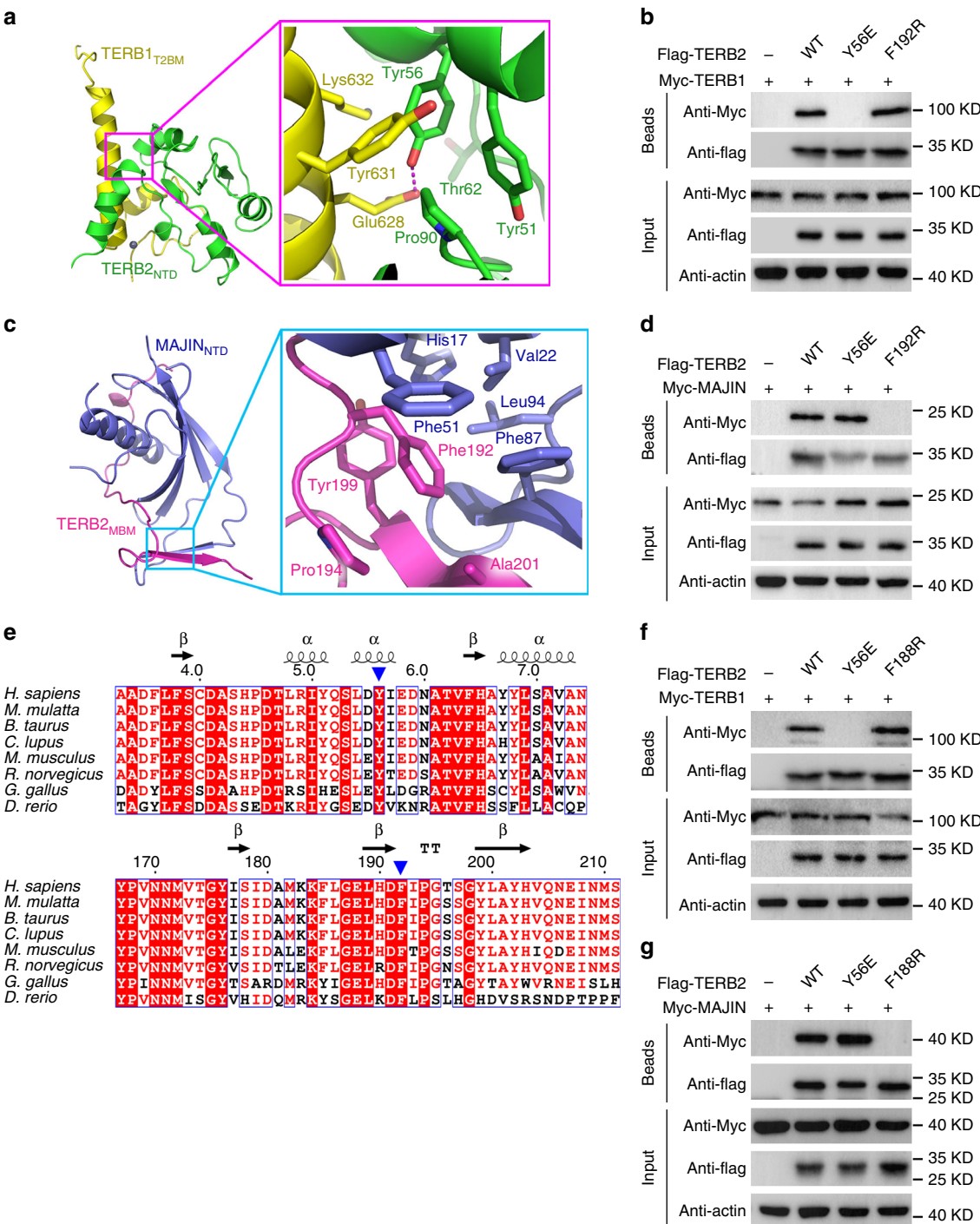

**Fig. 4** Mutational analyses of the TERB1–TERB2 and TERB2–MAJIN interactions. **a**, **c** Detailed interactions around Tyr56$^{TERB2}$ in the TERB1$_{T2BM}$–TERB2$_{NTD}$ complex (**a**) and around Phe192$^{TERB2}$ in the TERB2$_{MBM}$–MAJIN$_{NTD}$ complex (**c**). **b**, **d** Co-immunoprecipitation (Co-IP) of human TERB1 (**b**) or MAJIN (**d**) with wild-type (WT) or mutant Flag-tagged human TERB2 in 293T cells. The levels of each protein in the input and IP samples were analyzed by immunoblotting with the indicated antibodies. Input contains 5% of the input whole-cell lysate used for IPs. **e** Multiple sequence alignment of the N-terminal (top) and C-terminal (bottom) regions of TERB2 required for its association with TERB1 and MAJIN, respectively. Sequence positions are shown based on the human TERB2. The residues of Tyr56 (Try56 in mouse) and Phe192 (Phe188 in mouse) are indicated by blue arrowheads. **f**, **g** Co-IP of mouse TERB1 (**f**) or mouse MAJIN (**g**) with WT or mutant Flag-tagged mouse TERB2 in 293T cells. The levels of each protein in the input and IP samples were analyzed by immunoblotting with the indicated antibodies. Input contains 5% of the input whole-cell lysate used for IPs. Source data are provided as a Source Data file

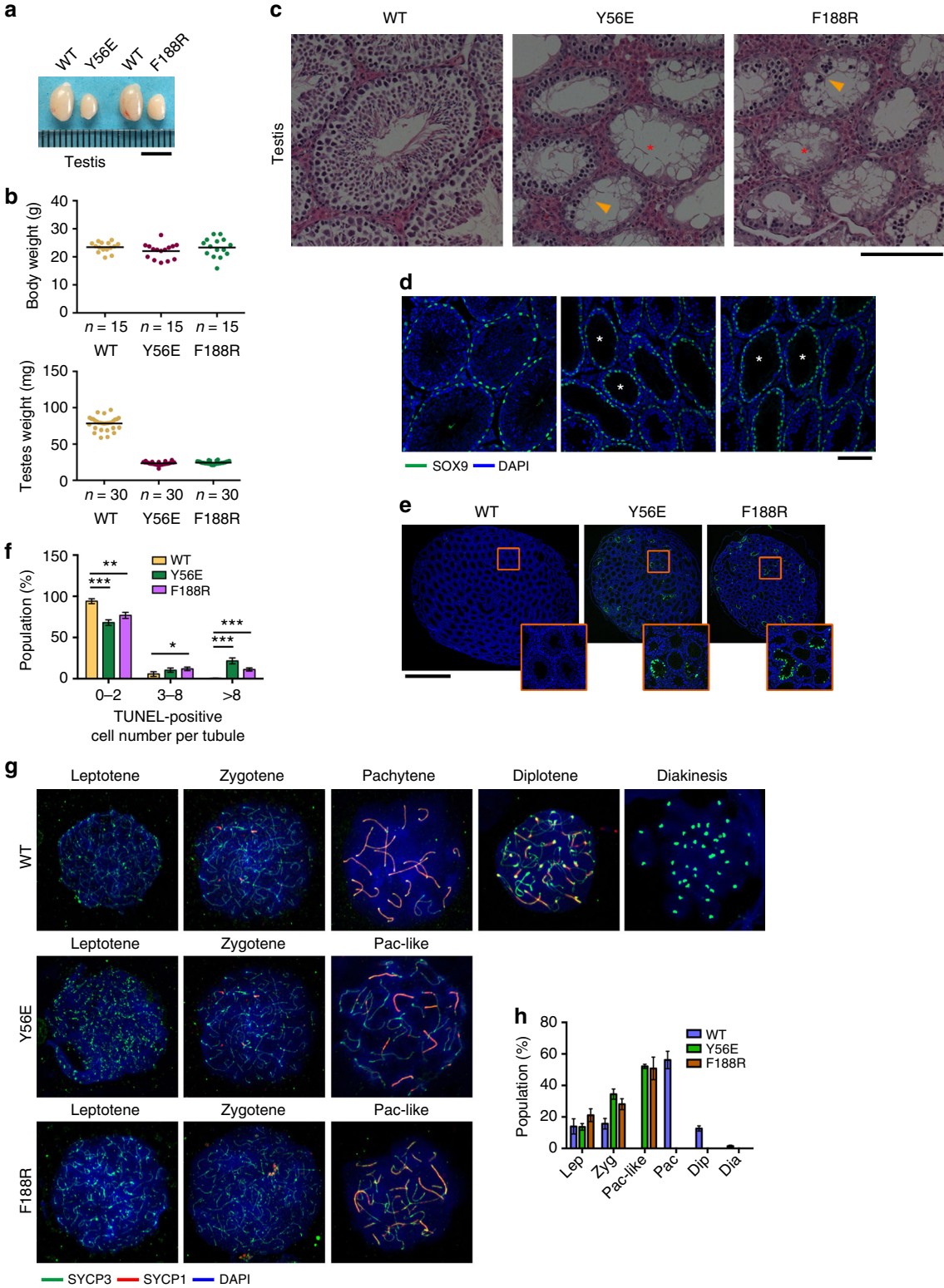

In both *Terb2*<sup>Y56E</sup> and *Terb2*<sup>F188R</sup> testes, although the presence of leptotene and zygotene spermatocytes was evident, meiotic progression was apparently arrested at a pachytene-like stage where autosomal SC formation was never completed but with some partially synapsed lateral elements as shown by the positive staining of SYCP1 (Fig. 5g, h). The prophase I arrest was further confirmed by the Hoechst 33342 staining profile of the *Terb2*<sup>Y56E</sup>

and *Terb2*<sup>F188R</sup> spermatocytes (Supplementary Fig. 3c). We never observed any *Terb2*<sup>Y56E</sup> and *Terb2*<sup>F188R</sup> spermatocytes that reached the mid-pachytene stage with typical WT SYCP3 and SYCP1 staining pattern along the entire autosomes (Fig. 5g, h).

IF analysis showed that TERB1, TERB2, and MAJIN were all localized at the ends of chromosomal axes in WT pachytene spermatocytes (Supplementary Fig. 4a–f)[12,13]. In contrast, TERB1

**Fig. 5** Disruption of the TERB1–TERB2–MAJIN complex results in defective spermatogenesis in mice. **a** Testes from 6-week-old wild-type, $Terb2^{Y56E}$, and $Terb2^{F188R}$ mice. Scale bars, 5 mm. **b** Body weight (top) and testis weight (bottom) of 6-week-old male mice of the indicated genotypes. **c** Hematoxylin and eosin-stained histological cross-sections of testes. Arrowheads indicate abnormal spermatocyte-like cells in mutant seminiferous tubules. Asterisks show vacuolated seminiferous lacking spermatozoa and spermatids. Scale bars, 100 μm. **d** Immunofluorescence (IF) staining for SOX9 (green) of testis sections from adult mice. DNA was stained by 4,6-diamidino-2-phenylindole (DAPI; blue). Asterisks show seminiferous containing a single layer of Sertoli cells and very few spermatogonial cells at the tubule periphery. Scale bars, 100 μm. **e** TUNEL staining of testis sections from 6-week-old mice. DNA was stained with DAPI (blue) and apoptosis was detected by IF terminal deoxinucleotidyl transferase-mediated dUTP-fluorescein nick end labeling (TUNEL) (green). Many TUNEL-positive cells were observed in both $Terb2^{Y56E}$ and $Terb2^{F188R}$ testes. Scale bars, 1 mm. **f** Population of TUNEL-positive tubules shown as mean and s.d. $n = 3$ mice of each indicated genotype. Statistical significances (*$P < 0.05$, **$P < 0.01$, ***$P < 0.001$) were assessed by two-tailed $t$ tests. **g** Representative spermatocyte nuclei from 6-week-old mice at different stages stained with DAPI and antibodies of SYCP3 and SYCP1. Pac-like Pachytene-like. **h** Quantification of the frequencies of meiotic stages shown in **g**. Data are from three independent experiments of different mice and error bars indicate s.d. More than 100 nuclei from each mouse were counted. Lep Leptotene, Zyg Zygotene, Pac-like Pachytene-like, Pac Pachytene, Dip Diplotene, Dia Diakinesis. Source data are provided as a Source Data file

and TERB2 but not MAJIN were present at chromosomal ends in $Terb2^{F188R}$ pachytene-like spermatocytes, and only TERB1 was retained at chromosomal ends in $Terb2^{Y56E}$ spermatocyte (Supplementary Fig. 4a–f). This result was in agreement with the in vitro Co-IP analysis in 293T cells (Fig. 4f, g), suggesting that the TERB2 mutations Y56E and F188R, respectively, disrupted the TERB1–TERB2 and the TERB2–MAJIN interactions in mutant spermatocyte in an expected manner.

Like their male siblings, homozygous $Terb2^{Y56E}$ and $Terb2^{F188R}$ female mice were also infertile (Supplementary Table 4). Although the mutant female mice exhibited similar body weight as their WT littermates, ovaries from mutant adults were greatly reduced in size as compared with those of WT mice (Supplementary Fig. 5a, b). Histological examination failed to identify any growing follicles in $Terb2^{Y56E}$ and $Terb2^{F188R}$ ovaries (Supplementary Fig. 5c). Moreover, both $Terb2^{Y56E}$ and $Terb2^{F188R}$ fetal oocytes displayed a complete meiotic arrest at pachytene-like stage similar to the mutant spermatocytes (Supplementary Fig. 5d, e). The most advanced fetal oocytes observed in the mutant mice exhibited abnormal pattern of SC formation with only partially synapsed lateral elements (Supplementary Fig. 5d, e). Taken together, we concluded that the intact TTM complex is required for gametogenesis and targeted disruption of the complex results in complete meiotic arrest in prophase I in both sexes.

**Disruption of the TTM complex induces disordered synapsis.** The presence of only partially synapsed lateral elements in the arrested mutant nuclei indicated a severe defect in synapsis (Fig. 5g, h). Moreover, the mutant spermatocytes exhibited a large increase of TRF1 foci and SYCP3 filaments compared to WT pachytene spermatocytes (Fig. 6a, b), suggestive of synapsis defects in both chromosome bodies and telomeric ends. To investigate the underlying mechanism for the synapsis defects in mutant mice, we performed fluorescence in situ hybridization (FISH) chromosome painting to examine the homologous association in spermatocytes. Majority of the arrested mutant spermatocytes exhibited separated chromosome signals (Supplementary Fig. 6a–d), consistent with the presence of large number of SYCP3 filaments (Fig. 6a). FISH experiments with probes to chromosomes 5 and 8 revealed similar results (Supplementary Fig. 6a–d), suggesting that defect of homologous association is not specific to particular chromosomes.

The partial synapsis in arrested mutant spermatocytes revealed by SYCP1 staining suggests a possibility of non-homologous synapsis formation (Fig. 5g). When only chromosome 5 was counted, most of the detected axial structures were asynapsed with no SYCP1 signals (~60%), some displayed self-pairing (~>20%), some exhibited non-homologous pairing (~5%), and

<20% showed likely homologous pairing (Fig. 6d). The most convincing evidence for the presence of heterologous chromosome synapsis in $Terb2$-mutant spermatocytes was the synapsis formation on a chromosome 5 that was clearly separated from its homolog revealed by chromosome painting (Fig. 6c). Notably, some chromosome 5 even homologously associated displayed partial synapsis (Fig. 6c). The aberrant synapsis in the mutant spermatocytes was confirmed by the stimulated emission depletion (STED) super-resolution microscopy. The mutant spermatocytes exhibited a lot of self-alignment, including self-synapsis of chromosomal termini (Fig. 6e). STED also revealed misaligned telomeres in some apparently fully synapsed axial structures (Fig. 6e, the zoom-in view). All these misalignments at chromosome ends explained the large number of telomere foci observed in $Terb2$-mutant spermatocytes (Fig. 6a, b). Taken together, our results indicated that disruption of the TTM complex causes failure of homologous association and disordered synapsis and thus leads to prophase I arrest.

**Disruption of the TTM complex impairs meiotic double-strand break (DSB) repair.** Meiotic DNA recombination that is initiated via programmed DNA DSBs is coupled with SC formation[23]. To further investigate the synapsis defects observed in $Terb2^{Y56E}$ and $Terb2^{F188R}$ male mice, we examined the effects of $Terb2$ mutations on homologous recombination-directed DSB repair. In both $Terb2^{Y56E}$ and $Terb2^{F188R}$ males, spermatocytes displayed positive γ-H2AX staining from leptotene through zygotene in a similar fashion as the WT spermatocytes (Fig. 7a). This result suggested that the initiation of the programmed DSBs is not affected in $Terb2$ mutant spermatocytes. It is noteworthy that, although reduced, γ-H2AX signals were still obvious in mutant pachytene-like spermatocytes, whereas the signal can only be observed in the X–Y body in WT pachytene nuclei (Fig. 7a), indicating that the programmed DSB repair process is severely impaired and no $Terb2^{Y56E}$ and $Terb2^{F188R}$ spermatocytes reach mid-pachytene stage (Fig. 7a, b). Next, we analyzed the location of DMC1, a protein involved in early repair of DSBs[24]. In WT zygotene spermatocytes, DMC1 assembles on the axial elements of chromosomes and gradually disappear towards pachynema (Fig. 7c, d). However, the $Terb2^{Y56E}$ and $Terb2^{F188R}$ spermatocytes exhibited persistent DMC1 foci (Fig. 7c, d), suggesting that the loading of DMC1 was not affected but subsequent process was arrested in mutant spermatocytes. To further examine the defect of DSB repair in mutant mice, we analyzed the distributions of MSH4, a marker of intermediate steps in recombination that forms discrete foci along aligned chromosomes[25], and of MLH1, a key player in the last step of recombination leading to the formation of crossovers[26]. We found that majority of SYCP3 filaments lacked MSH4 foci in the mutant pachytene-like

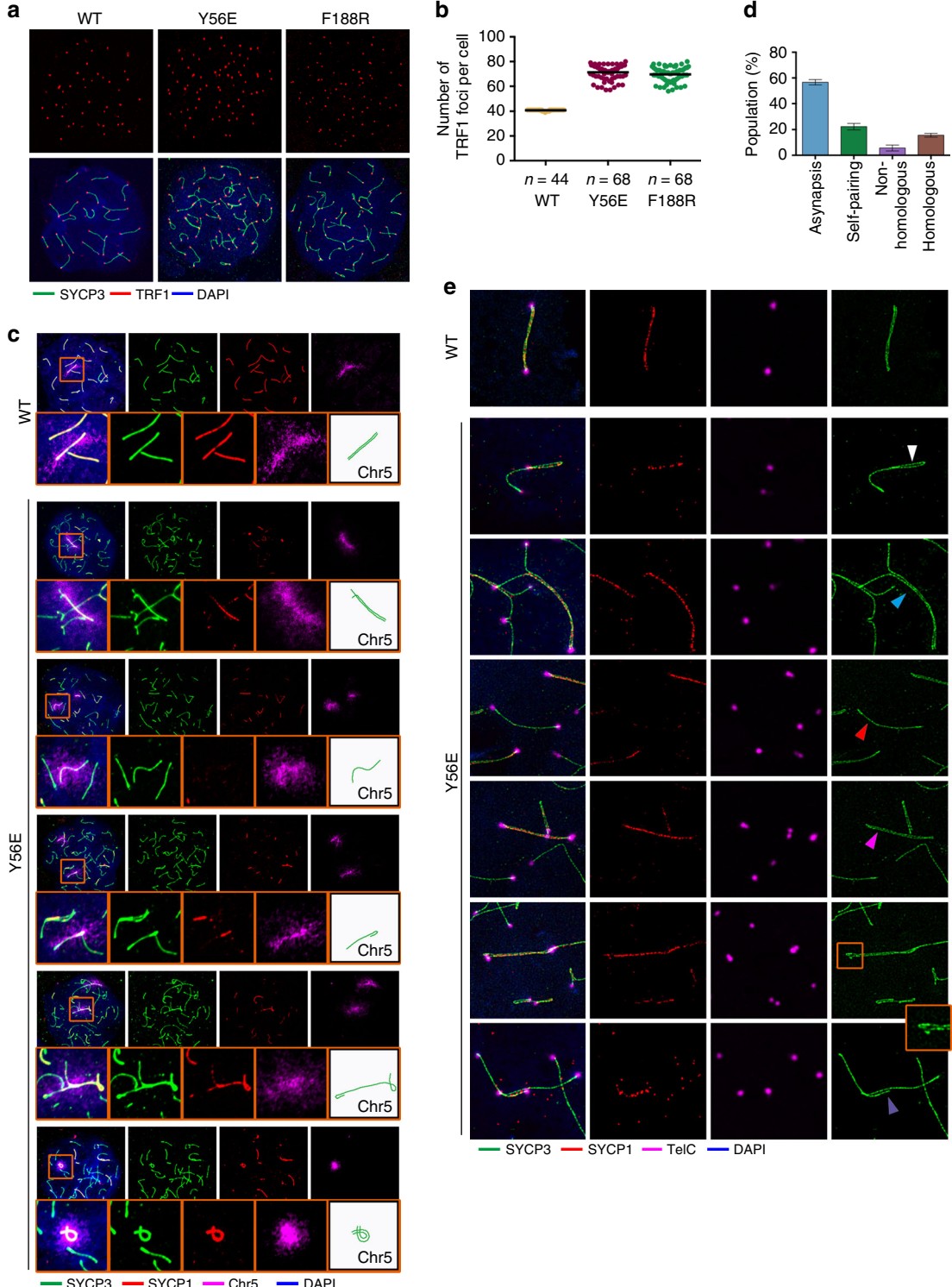

spermatocytes, suggestive of a great reduction of DNA–DNA interaction between homologous chromosomes (Fig. 7e, f). This is consistent with the result of chromosome painting assays that most chromosomes are spatially separated from their homologs (Supplementary Fig. 6a–d). As expected, IF staining revealed no MLH1 foci in the mutant spermatocytes (Fig. 7g, h), suggesting a complete absence of crossover formation between homologous chromosomes. Collectively, our data demonstrated that disruption of the TTM complex causes severe impairment of homologous recombination and persistent DSBs.

**Disruption of the TTM complex abolishes telomere–NE linkage.** Next, we analyzed telomere distribution patterns in structurally preserved spermatocytes by staining of SYCP3, the NE marker Lamin B, and the telomeric DNA probe. IF-FISH assays revealed that majority of telomere foci in WT nuclei are localized to the nuclear periphery (Fig. 8a). In contrast, both the *Terb2*<sup>Y56E</sup> and *Terb2*<sup>F188R</sup> spermatocytes exhibited scattered telomeric signals within the nuclei, suggesting that most of telomeres in *Terb2* mutant spermatocytes did not attach to the NE (Fig. 8a, b). Similar conclusion was obtained when testis sections were used

**Fig. 6** Disruption of the TERB1–TERB2–MAJIN complex results in aberrant synapsis. **a** Chromosome spreads from adult spermatocytes were immunostained with anti-SYCP3 (green) and anti-TRF1 (red). **b** Quantification of TRF1 foci number per nucleus ($n > 40$ cells for each genotype) shown in **a**. Data are from three independent experiments of three different mice. Note that there are 40 chromosomes (2N) in murine cells and the number of telomere foci ranges from 80 (fully unpaired) to 41 (fully paired) in wild-type spermatocytes. **c** Fluorescent images of spermatocyte spreads labeled with anti-SYCP3 (green), anti-SYCP1 (red), and chromosome 5 painting (magenta). DNA was stained by 4,6-diamidino-2-phenylindole. Representative images of Y56E spreads show various disordered synapsis of chromosome 5. Insets are enlarged figures of the framed regions. SYCP3 filaments of chromosome 5 were indicated by sketches. **d** Quantification of the frequencies of different status of chromosome 5. Error bars indicate s.d. of three independent experiments with different mice. For each mouse, >50 pachytene-like spreads were counted for quantification. **e** Immunofluorescence–fluorescence in situ hybridization (FISH) analysis by stimulated emission depletion of telomere (FISH), SYCP1, and SYCP3 on spermatocyte spreads from adult testes. The white arrowhead indicates self-pairing, the light blue arrowhead shows a Y-shaped partial pairing, the red arrowhead points to a non-pairing chromosome, the magenta arrowhead indicates an apparently fully paired chromosome, the zoom-in view shows the bend on the top of the apparently fully paired chromosomes, and the purple arrowhead shows a bridge-shaped partial pairing. Source data are provided as a Source Data file

for IF-FISH analysis (Supplementary Fig. 7). To test whether some synapsed chromosomes in *Terb2* mutant spermatocytes may properly attach to the NE, we prepared testis tissue for in situ electron microscopic analysis, by which stretches of SC and sites of telomere–NE attachment can easily be detected. As anticipated, synapsed stretches of SC attached to the NE were clearly evident in WT spermatocytes but not in *Terb2*$^{Y56E}$ or *Terb2*$^{F188R}$ nuclei (Fig. 8c, d), further confirming that disruption of the TTM complex abolishes the telomere–NE attachment.

SUN1 resides on the NE and is required for the telomere-LINC complex connection[27]. Notably, no SUN1 foci were detected at chromosomal ends on *Terb2*$^{Y56E}$ or *Terb2*$^{F188R}$ spermatocyte spreads (Fig. 8e, f). This result suggested that the telomere–NE attachment mediated by the TTM complex is necessary for the telomere-LINC complex connection. On the other hand, although there were still telomere foci at the nuclear periphery, we observed an increased number of internal telomere signals in *Sun1*$^{-/-}$ spermatocytes compared to WT (Fig. 8a, b)[28], suggesting that, despite SUN1 is not a prerequisite for the telomere–NE association, it plays an important role in the stable attachment of telomeres to the NE. This is consistent with the reduced frequency of telomere–NE junction observed by electron microscopic analysis in *Sun1*$^{-/-}$ spermatocytes (Fig. 8c, d). Together, we conclude that both the TTM complex and SUN1 (the LINC complex) contribute to the stable telomere–NE association.

**Disruption of TTM complex causes Speedy A congregation at NE.** CDK2 was previously shown to localize at telomeres, late recombination nodules, and the sex body in prophase I nuclei[29], playing an essential role in meiotic prophase I progression[19,30,31]. Speedy/RINGO A, a noncanonical activator of CDK2, was recently identified as a key regulator for CDK2 recruitment to meiotic telomeres[17,18]. Knockout studies showed that both CDK2 and Speedy A contribute to efficient NE attachment of telomeres[17–19]. This prompted us to examine the distribution of Speedy A and CDK2 in the absence of the telomere–NE attachment in *Terb2*$^{Y56E}$ and *Terb2*$^{F188R}$ spermatocytes. Consistent with previous studies[17,18], WT zygotene and pachytene spermatocytes displayed clear staining of both Speedy A and CDK2 at telomeres (Fig. 9a, c)[14,15]. In sharp contrast, telomere-bound Speedy A and CDK2 were greatly reduced in *Terb2*$^{Y56E}$ and *Terb2*$^{F188R}$ zygotene spermatocytes (Fig. 9a–d), suggesting that the telomere–NE attachment is important for the stable association of Speedy A and CDK2 with telomeres. Strikingly, however, many *Terb2* mutant spermatocytes exhibited sustained congregation of Speedy A at the NE, suggesting that a fraction of Speedy A associate with the NE independent of the telomere–NE attachment (Fig. 10a, b). Moreover, this congregation of Speedy A at the NE was not detectable in *Sun1*$^{-/-}$ spermatocytes, indicating that Speedy A congregation at the NE is SUN1 dependent

(Fig. 10a, b). Collectively, our data revealed two pools of Speedy A in the absence of telomere–NE attachment, one on the NE and the other at telomeres. The substantial reduction of telomere-bound Speedy A and CDK2 in *Terb2* mutant spermatocytes suggested that the telomere–NE attachment promotes the association of Speedy A and CDK2 to the telomeres.

To investigate the mechanism of Speedy A congregation at the NE when the TTM complex is disrupted, we next analyzed SUN1 distribution in *Terb2*$^{Y56E}$ and *Terb2*$^{F188R}$ spermatocytes. In WT spermatocytes, telomeres, led by the LINC complex, cluster together to form the bouquet during a short time window in late zygotene stage and disperse along the NE after early pachytene[12,13]. Our IF-FISH data of testis sections revealed ~4% of WT nuclei showing congregated SUN1 staining (Fig. 10c, d), consistent with the conclusion that telomere bouquet formation is a transient event[12,13]. Notably, congregation of SUN1 at the NE was also observed in both *Terb2*$^{Y56E}$ and *Terb2*$^{F188R}$ nuclei, suggesting that SUN1 moving and clustering on the NE in early prophase I is independent of the telomere–NE connection (Fig. 10c). Strikingly, similar to the sustained congregation of Speedy A (~38%) at the NE, the population of nuclei displaying congregation of SUN1 in mutant mice was also much higher (~35%) compared to that in WT spermatocytes, suggesting that the dispersion of SUN1 and Speedy A is severely delayed or arrested in mutant nuclei (Fig. 10a–d). Together, we concluded that, in *Terb2* mutant spermatocytes, a pool of Speedy A that is not associated with telomeres is recruited to and congregated at the NE in a SUN1-dependent manner.

## Discussion

During meiotic prophase I, telomeres attach to and move on the NE, leading chromosome movements, a dynamic process essential for meiosis. SUN1 and MAJIN are two membrane proteins that provide binding sites on the inner surface of the NE for telomere attachment. However, it has been unclear how these two proteins and their binding partners cooperate together to establish the stable attachment of telomeres to the NE. In this study, we determined the crystal structures of the human TERB1$_{T2BM}$–TERB2$_{NTD}$ and TERB2$_{MBM}$–MAJIN$_{NTD}$ complexes and helped reveal at molecular level how these proteins establish the telomere–NE connection. We generated knock-in mice with TERB2 mutations that specifically disrupt its interaction with TERB1 (*Terb2*$^{Y56E}$) or MAJIN (*Terb2*$^{F188R}$). Both mutations resulted in complete disruption of the telomere–NE attachment in mouse spermatocytes (Fig. 8a–f), suggesting that the TTM complex is essential for telomere–NE connection. In addition, we also examined the meiosis progress in *Sun1*$^{-/-}$ mice. Consistent with previous data that SUN1 plays an important role in telomere–NE association[28], we found that the telomere–NE attachment was partially disrupted by the depletion of SUN1 (Fig. 8a, b). A previous study revealed a direct connection between the TTM complex and the LINC complex through an

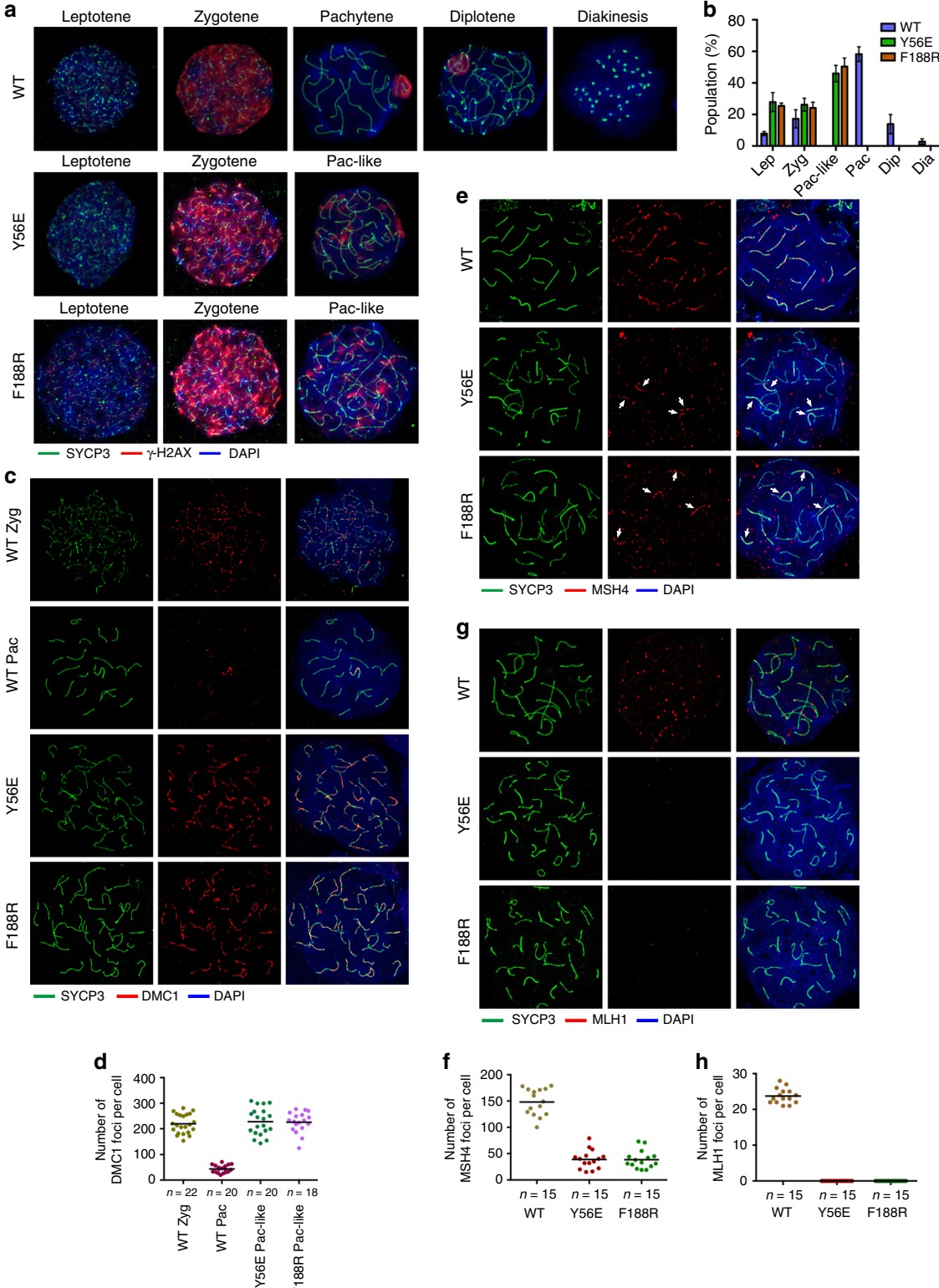

interaction between SUN1 and TERB1[15]. However, disruption of the interactions within the TTM complex results in complete abolishment of the telomere–NE attachment and depletion of SUN1 from telomeres (Fig. 8a–f), indicating that the SUN1–TERB1 interaction might be weak and the stable telomere–LINC complex connection requires additional interaction(s) between telomeres and SUN1. Taken together, we propose that the telomere–TRF1–TERB1–TERB2–MAJIN–NE interaction network and the telomere–LINC complex connection are likely two separate but cooperative pathways to stably recruit telomeres to the NE in meiosis prophase I. To fully understand the telomere–NE connection, further study is needed to reveal the molecular mechanism of the telomere–LINC complex connection.

The telomere–LINC connection is suggested to be a prerequisite for meiotic chromosome movements and bouquet formation that is required for efficient progression of prophase I[10].

**Fig. 7** Recombination and double-strand break repair was severely impaired in mutant spermatocytes. **a** Representative spermatocyte nuclei from 6-week-old mice at different indicated stages immunostained with SYCP3 (green) and γ-H2AX (red). Pac-like Pachytene-like stage. **b** Quantification of the frequencies of meiotic stages shown in **a**. Data are from three independent experiments of different mice and error bars indicate s.d. More than 100 nuclei from each mouse were counted. Lep Leptotene, Zyg Zygotene, Pac-like Pachytene-like, Pac Pachytene, Dip Diplotene, Dia Diakinesis. **c** Immunofluorescence staining of spermatocyte chromosome spreads for SYCP3 (green) and DMC1 (red). **d** Quantification of the number of DMC1 foci on the SYCP3 filaments per nucleus. **e** Immunofluorescence staining of spermatocyte chromosome spreads for SYCP3 (green) and MSH4 (red). White arrows indicate chromosomes with MSH4 foci in the mutant nuclei. **f** Quantification of the number of MSH4 foci on the SYCP3 filaments per nucleus shown in **e**. **g** Immunofluorescence staining of spermatocyte chromosome spreads for SYCP3 (green) and MLH1 (red). **h** Quantification of the number of MLH1 foci on the SYCP3 filaments per nucleus shown in **g**. DNA were stained with 4,6-diamidino-2-phenylindole (**a**, **c**, **e**, **g**). Source data are provided as a Source Data file

Telomere localization of Speedy A and its potential role in regulating telomere–NE attachment in meiotic prophase I prompted us to examine its distribution when telomeres are depleted from the NE[17,18]. We observed two pools of Speedy A, one with telomeres in the nucleoplasm and the other on the NE, when telomeres are depleted from the NE by mutations of TERB2 (Figs. 9a and 10a). The presence of telomere-associated Speedy A is consistent with a recent study, which identified a direct interaction between Speedy A and TRF1 (Fig. 9a, b)[32]. However, a substantial reduction of Speedy A at telomeres in both $Terb2^{Y56E}$ and $Terb2^{F188R}$ spermatocytes compared to WT indicated that the telomere–NE attachment promotes the recruitment of Speedy A to telomeres (Fig. 9a, b). On the other hand, the NE-associated Speedy A that appears as congregations without telomere signals is dependent on the presence of SUN1 (Fig. 10a, b), suggesting that this pool of Speedy A moves and congregates in a LINC complex-dependent manner. Taken together, we speculate that speedy A is likely capable of interacting with both telomeres and the LINC complex and thus might function as the missing linkage between telomeres and the LINC complex during prophase I, stabilizing the telomere–NE connection. It will be of great interest to test this hypothesis to fully understand the mechanisms of stable telomere–NE connection and telomere movement along the NE driven by the LINC complex.

The knock-in mice with specific and accurate disruption of the TERB1–TERB2 or the TERB2–MAJIN interactions enable us to systematically assess the consequent meiosis defects caused by the disruption of the telomere–NE attachment. During WT early prophase I, the conserved programmed DSB repair triggers a genome-wide search for DNA sequence homology[23]. In both $Terb2^{Y56E}$ and $Terb2^{F188R}$ spermatocytes, the γ-H2AX signal was detected at leptotene and zygotene stages similar to that in WT spermatocytes (Fig. 7a, b), indicating the normal initiation of the programmed DSBs. Consistently, loading of DMC1 on the chromosomes suggests that DNA–RAD51 complexes were formed in mutant spermatocytes (Fig. 7c, d). However, several lines of evidence suggest that the subsequent DSB repair process was severely impaired. First, the DMC1 signal is persistently remained on the chromosomes in mutant pachytene-like spermatocytes (Fig. 7c, d). Second, majority of homologous chromosomes are not well aligned to each other (Fig. 6a, b) or not even associated to each other as revealed by the chromosome painting assays (Supplementary Fig. 6a–6d). Third, in mutant spermatocytes, no MLH1 foci were observed on the chromosomes, suggestive of no crossover formation (Fig. 7g, h). Collectively, we conclude that disruption of the telomere–NE connection does not interfere the initiation of the programmed DSBs, but it interrupts the subsequent homologous recombination-directed repair process. Strikingly, besides the asynapsed chromosomes, we observed a substantial level of self-pairing and non-homologous pairing of chromosomes (Fig. 6c, d), suggesting that disruption of telomere–NE attachment results in disordered synapsis. It has been suggested that telomere-led meiotic chromosome movements facilitate chromosome pairing[10,14] and that the

telomere–LINC connection is a prerequisite for these chromosome movements[10]. Disruption of the telomere–NE attachment prevents telomere–LINC complex connection and therefore disables meiotic chromosome movements. We speculate that lack of chromosome movements may induce incorrect heterologous pairing, resulting in the observed heterologous synapsis in $Terb2$ mutant spermatocytes. Future studies will be needed to fully understand how telomere-led chromosome movements influence homologous pairing in meiosis.

## Methods

**Protein expression and purification.** Human TERB1$_{587–659}$ and TERB2$_{2–116}$ were co-expressed in *Escherichia coli* BL21 CodonPlus (DE3) cells (Stratagene) using a modified pRSFDuet vector with a 6×His tag fused at the N-terminus of TERB2. The primers used for cloning are as follows. TERB1(587–659)-F: ATT GGA TCC ATG CTG ACC TAC CGT TGC, TERB1(587–659)-R: TTT CTC GAG TTA TTA AGA TTC GTT AGA CAG ACG CTG A, TERB2(2–116)-F: AAT TGG ATC CTT TCA AGG GCA GCG CGG TTG G, TERB2(2–116)-R: AAT TCT CGA GTT ACT TTT CTA TCA GAA AAT GTT GGT C. After induction for 19 h with 0.1 mM IPTG at 18 °C, cells were harvested by centrifugation, resuspended in lysis buffer (50 mM Tris-HCl pH 8.5, 500 mM NaCl, 400 mM Urea, 10% Glycerol, 1 mM PMSF, 5 mM benzamidine, 1 μg/mL leupeptin, 1 μg/mL pepstatin, and 0.025 Unit supernuclease), and lysed by sonication. Cell lysate was ultracentrifuged and supernatant was incubated with Ni-NTA agarose beads (Qiagen) at 4 °C with rocking for 1 h. After washing, the bound proteins were eluted by 250 mM imidazole. Protein sample was further purified by Mono-Q and gel-filtration chromatography equilibrated with the buffer (25 mM Tris-HCl pH 6.6, 150 mM NaCl). The TERB1$_{587–659}$–TERB2$_{2–116}$ complex was concentrated to 30 mg/mL and stored at −80 °C. Human TERB2$_{119–220}$ and MAJIN$_{2–149}$ were co-expressed and purified from *E. coli* BL21 CodonPlus (DE3) cells (Stratagene) using the same strategy as for the TERB1$_{587–659}$–TERB2$_{2–116}$ complex, except that the buffers for lysis and chromatography had a pH value of 7.5. Primers used for cloning are as follows. TERB2(119-220)-F: ACA GGA TCC GAA GTA ACA CCA AAT GAA ATA AAG ACC, TERB2(119-220)-R: TAG CTC GAG CTA TTT CCT CTT CAA TTT GTT TTT TAT AG, MAJIN(2-149)-F: AAT TGG ATC CAG TTT AAA ACC TTT TAC CTA CC, MAJIN(2-149)-R: AAT TCT CGA GTT AGC TGG GCT CGT CCA TGT G. The TERB2$_{119–220}$–MAJIN$_{2–149}$ complex was concentrated to 9 mg/mL and stored at −80 °C.

**Crystallization and structure determination.** Crystals of the native TERB1$_{587–659}$–TERB2$_{2–116}$ complex (30 mg/mL) were grown by sitting-drop vapor diffusion at 18 °C. The precipitant solution consisted of 0.1 M Bicine pH 9.0, 2% (v/v) 1,4-Dioxane, and 4% (w/v) Polyethylene glycol 20,000. For preparation of a mercury derivative, the dehydrated crystals were soaked in the dehydration buffer with 0.1 mM HgBr for 1 h. All of the crystals were gradually transferred into a harvesting solution (1 M Bicine pH 9.0, 2% (v/v) 1,4-Dioxane, 15% (w/v) Polyethylene glycol 20,000, 25% (v/v) Glycerol) followed by flash-freezing in liquid nitrogen for storage. Datasets were collected under cryogenic conditions (100 K) at the Shanghai Synchrotron Radiation Facility (SSRF) beamlines BL18U1 and BL19U1. A 3.9-Å Hg-SAD dataset of the TERB1$_{587–659}$–TERB2$_{2–116}$ complex was collected at the wavelength of 0.97853 Å and was processed by HKL3000[33]. One mercury atom was located and refined, and the single-wavelength anomalous diffraction data phases were calculated using Phenix[34]. The initial SAD map was substantially improved by solvent flattening. An initial model was automatically built into the modified experimental electron density map. The model was then refined using a 3.3-Å native dataset with manual building in Coot[35].

Crystals of SeMet-labeled TERB2$_{119–220}$–MAJIN$_{2–149}$ complex (9 mg/mL) were grown by sitting-drop vapor diffusion at 18 °C. The precipitant solution consisted of 0.2 M Potassium thiocyanate, 20% (w/v) Polyethylene glycol 3350, pH 6.8. Crystals were gradually transferred into a harvesting solution (100 mM HEPES pH 6.8, 0.2 M potassium thiocyanate, 22% (w/v) Polyethylene glycol 3350, 25% (v/v) Glycerol) followed by flash-freezing in liquid nitrogen for storage. Datasets were collected under cryogenic conditions (100 K) at the SSRF beamlines BL18U1 and

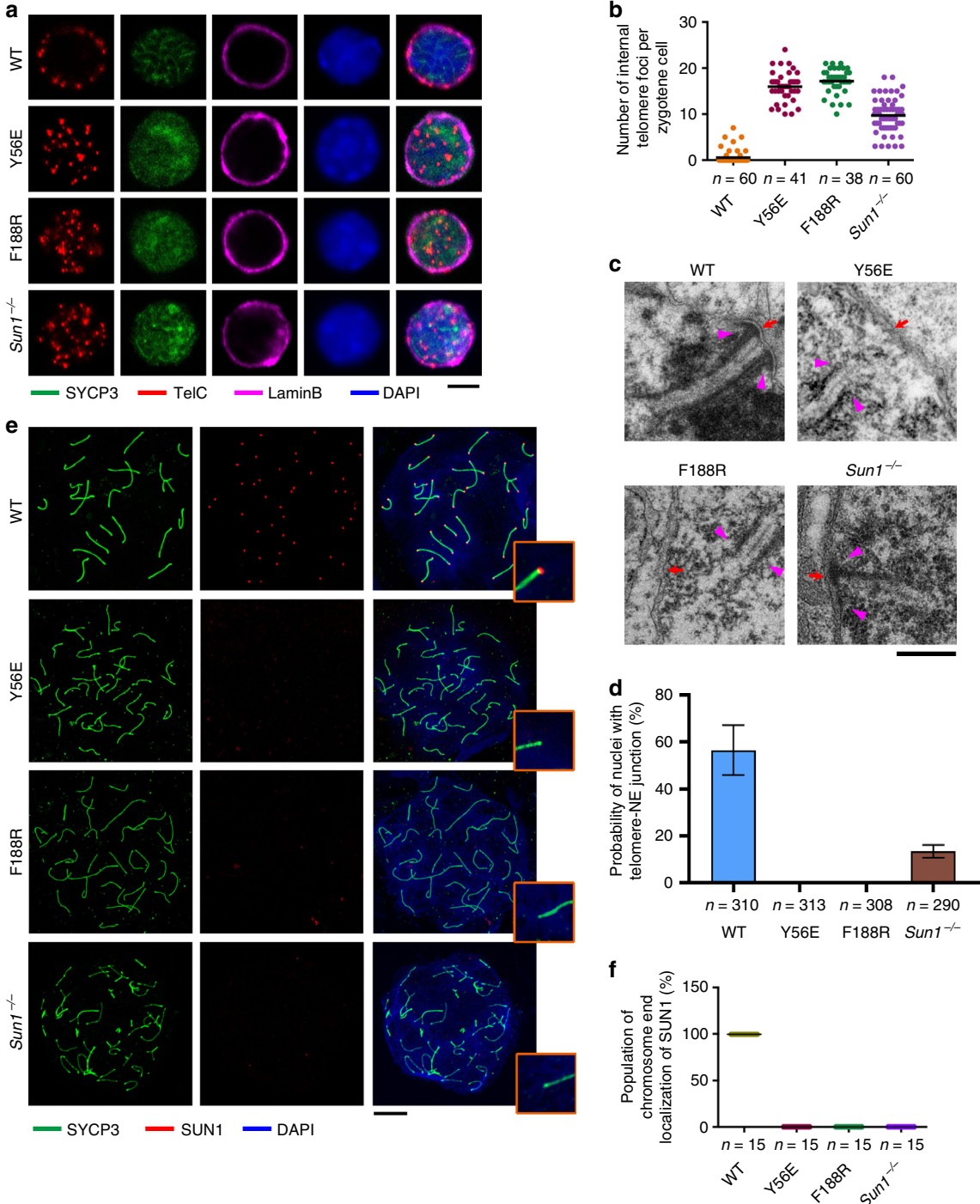

**Fig. 8** Intact TERB1–TERB2–MAJIN complex is required for telomere–nuclear envelope (NE) attachment. **a** Equator images of structurally preserved zygotene spermatocytes from wild-type (WT), *Terb2^{Y56E}*, *Terb2^{F188R}*, and *Sun1^{−/−}* immunostained for SYCP3 and Lamin B and subjected to fluorescence in situ hybridization of telomeric DNA probe (TelC). DNA was stained by 4,6-diamidino-2-phenylindole (DAPI). Scale bars, 5 μm. **b** Quantification of internal telomere foci number shown in **a**. **c** Electron micrographs showing telomeres (magenta arrowheads) and nuclear membrane (red arrows) in WT pachytene and mutant pachytene-like spermatocytes. Scale bars, 500 nm. **d** Populations of nuclei from spermatocyte sections showing at least one telomere–NE attachment. Only nuclei with stretches of synaptonemal complex (SC) observed were counted for the quantification. Data are from three independent experiments with different mice and error bars indicate s.d. About 100 nuclei containing SC stretches from each mouse were counted for quantification. *n* is the total number of nuclei with SC stretches used for counting. **e** Chromosome spreads from adult spermatocytes were immunostained with anti-SYCP3 (green) and anti-SUN1 (red). DNA was stained by DAPI. The zoom-in views show one chromosome end from each spread. Scale bars, 10 μm. **f** Dot blot showing population of chromosome ends with SUN1 foci in one nucleus. Source data are provided as a Source Data file

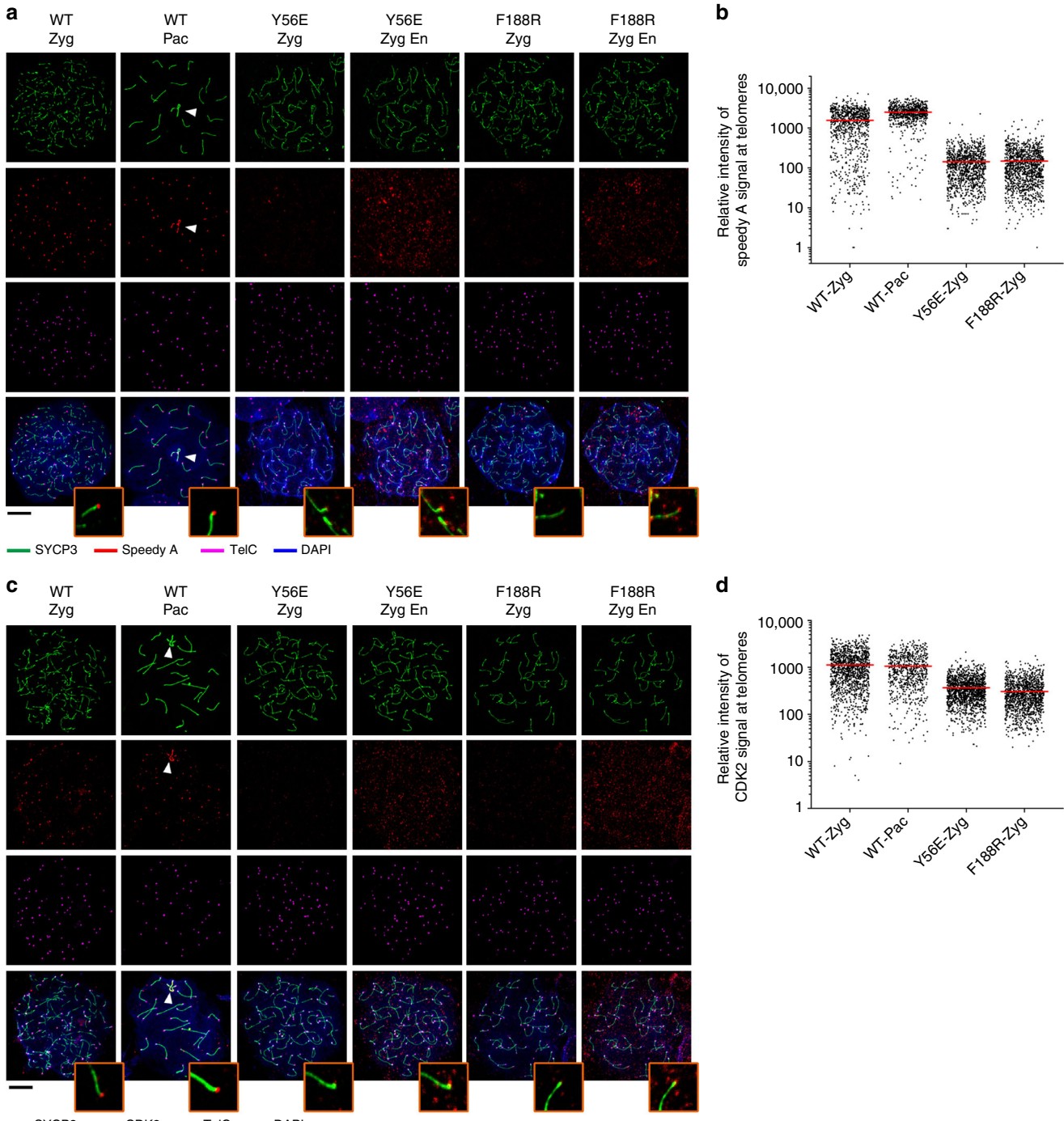

**Fig. 9** Telomere localization of Speedy A and CDK2 is reduced in mutant nuclei. **a** Immunofluorescence–fluorescence in situ hybridization (IF-FISH) images of chromosome spreads from adult spermatocytes of the indicated genotypes. Chromosome spreads were immunostained for SYCP3 (green) and Speedy A (red) and subjected to telomere FISH (TelC-FISH) (magenta). DNA was stained by 4,6-diamidino-2-phenylindole (DAPI; blue). Zyg Zygotene, Pac Pachytene, En showing enhanced signals (red) of Speedy A with a lower cutoff. White arrowheads indicate sex chromosomes. Scale bars, 10 μm. **b** Quantification of the relative intensity of Speedy A foci at telomeres in wild-type (WT) or *Terb2*-mutant spermatocytes in **a**. **c** IF-FISH images of chromosome spreads from adult spermatocytes of the indicated genotypes. Chromosome spreads were immunostained for SYCP3 (green) and CDK2 (red) and subjected to telomere FISH (TelC-FISH) (magenta). DNA was stained by DAPI (blue). Zyg Zygotene, Pac Pachytene, En showing enhanced signals (red) of CDK2 with a lower cutoff. White arrowheads indicate sex chromosomes. Scale bars, 10 μm. **d** Quantification of the relative intensity of CDK2 foci at telomeres in WT or *Terb2*-mutant spermatocytes in **c**. Source data are provided as a Source Data file

BL19U1. A 2.9-Å Se-SAD dataset of the $\mathrm{TERB2}_{119-220}$–$\mathrm{MAJIN}_{2-149}$ complex was collected at the Se peak wavelength (0.97776 Å) and was processed by HKL3000[33]. Three selenium atoms were located and refined, and the single-wavelength anomalous diffraction data phases were calculated using Phenix[34]. The initial SAD map was substantially improved by solvent flattening. An initial model was automatically built into the modified experimental electron density map. The model was then refined using Phenix[34], together with manual building in Coot[35]. All the structural figures were generated using PyMOL[36].

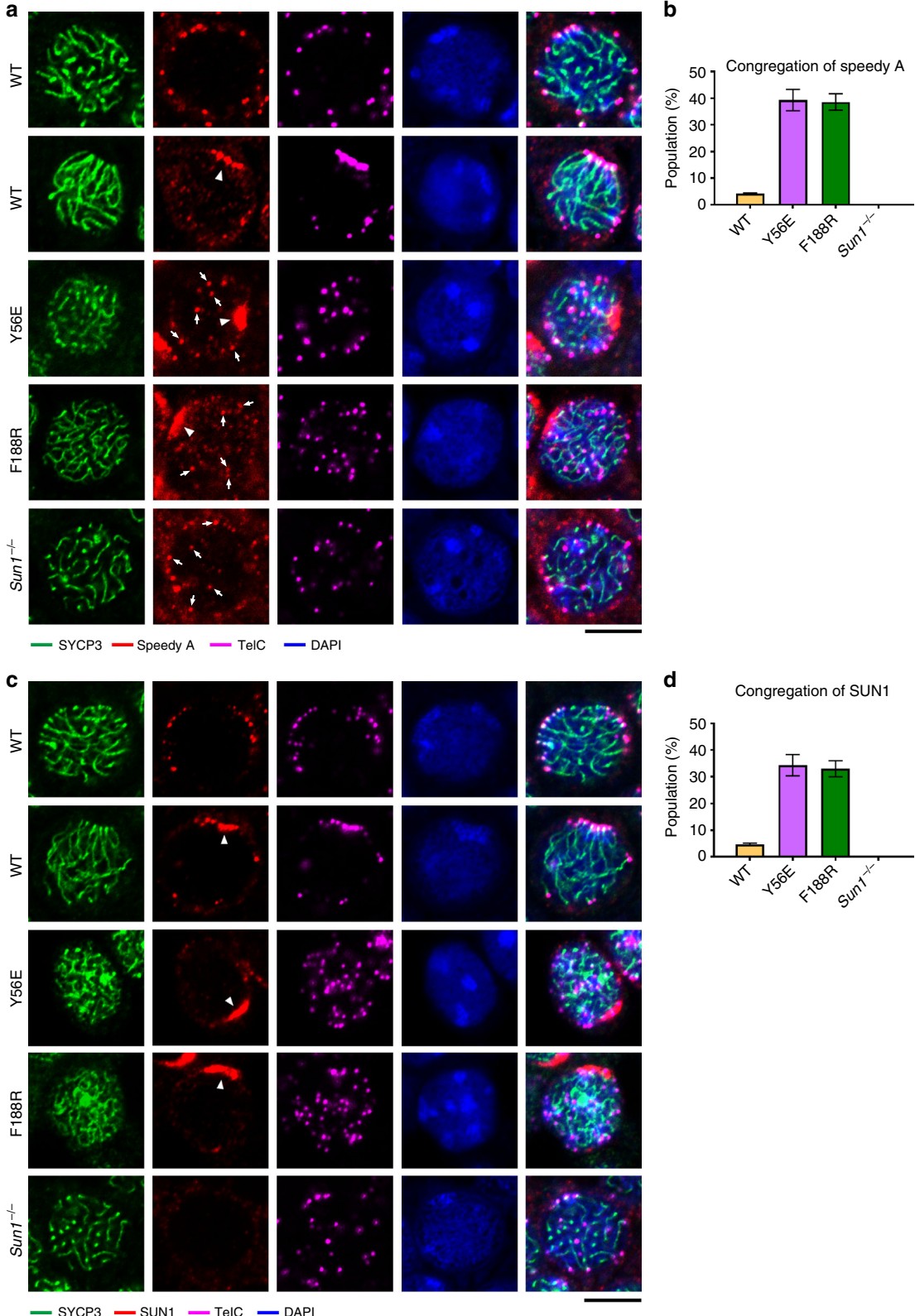

**Cell culture**. HEK293T and U-2 OS cells were obtained from the Cell Bank of Type Culture Collection of Chinese Academy of Sciences. Cells were cultured at 37 °C under 5% $CO_2$ in Dulbecco's modified Eagle's medium (DMEM) supplemented with 10% fetal bovine serum (FBS) (Hyclone). No mycoplasma contaminations were detected in these cells.

**BiFC assay**. cDNA fragments of human TERB1 and TERB2 were constructed into pLVX-IRES-Puro vectors (Clontech) and fused at their C termini with a Venus protein N-terminus (1–154) (-VN) or C-terminus (155–238) (-VC). Plasmids were transfected into HEK293T cells with the help plasmids pCMV-dR8.9 and pCMV-

**Fig. 10** Speedy A and SUN1 exhibit congregation in *Terb2* mutant spermatocytes. **a**, **c** Immunofluorescence–fluorescence in situ hybridization (FISH) showing distribution of Speedy A (**a**) and SUN1 (**c**) in spermatocytes. Sections of 2-week-old testes of different genotypes as indicated were immunostained for SYCP3 and Speedy A (**a**) or SYCP3 and SUN1 (**c**) and subjected to telomere FISH (TelC). DNA was stained by 4,6-diamidino-2-phenylindole. White arrowheads indicate congregation of Speedy A (**a**) or SUN1 (**c**). White arrows in **a** indicate Speedy A signals co-localized with telomeres. Scale bars, 5 μm. **b** Population of nuclei that display Speedy A congregation in **a**. Data are from three independent experiments with different mice and error bars indicate s.d. **d** Population of nuclei that display SUN1 congregation in **c**. Data are from three independent experiments with different mice and error bars indicate s.d. For each experiment, >100 spermatocytes from each indicated genotype were counted for the frequencies of nuclei showing congregation of Speedy A (**b**) or SUN1 (**d**). Source data are provided as a Source Data file

VSV-G with X-tremeGENE HP DNA Transfection Reagent (Roche) according to the manual. The lentiviral supernatants were harvested 36 h later and filtered by 0.22-μm filter (Millipore). Lentiviruses for -VN and -VC protein expression were used for co-infection of HEK293T cells. After 36 h, infected HEK293T cells were subjected to FACS analysis for Venus signals.

**Animals.** Mice were housed under controlled environmental conditions with free access to water and food. Experimental protocols were approved by the regional ethical committee of the National Center for Protein Science Shanghai (approval #SIBCB-S342-1510-043). Every effort was made to minimize and refine the experiments to avoid animal suffering.

**Generation of *Terb2*-mutant mice and *Sun1*[−/−] mice.** The mutant mice were generated by CRISPR/Cas9-mediated genome engineering[37]. Briefly, a single guide RNA (sgRNA: CAT TAT CTT CTA TGT ACT CG for *Terb2*[Y56E], GCC AAA TAT CCT GAG CTA CC for *Terb2*[F188R], and AGA CCA AAG TGG ACC ACT TC for *Sun1*[−/−]) was cloned into a sgRNA vector and was then used as a template to synthesize sgRNAs using the MEGAshortscript T7 Kit (Life Technologies). Cas9 mRNA was in vitro transcribed from plasmid MLM3613 (Addgene #42251) using the mMESSAGE mMACHINE T7 Ultra Kit (Life Technologies). For microinjection, Cas9 mRNA (100 ng/μL) was mixed with sgRNA (50 ng/μL) and 100 ng/μL donor oligos (CAT TCT TGT TTA TTT CAT AAC CAT TGA CTT CCT TTC TTC TAG AAT ATA TCA GAG CCT CGA AGA GAT AGA AGA TAA TGC TAC AGT TTT TCA TGC CTA CTA TCT TGC TGC AAT AGC TAA TAC TGA AA for *Terb2*[Y56E], CAT AGG CTA TAT ATC AAT TGA TGC CTT GGA GAA ATT CCT TGG AGA ATT ACA TGA CAG GAC TCC TGG ATC CTC AGG ATA TTT GGC ATA TCA TAT TCA AGA TGA AAT TAA TAT GTC TGC TAT for *Terb2*[F188R] and no donor for *Sun1*[−/−]) and microinjected into the cytoplasm of fertilized eggs collected from C57BL/6. The injected zygotes were cultured in M2 medium at 37 °C under 5% $CO_2$ for overnight. The embryos that had reached two-cell stage of development were implanted into the oviducts of pseudo-pregnant ICR foster mothers. The mice born from the foster mothers were genotyped by PCR of tail biopsy. For *Terb2*[Y56E] and *Terb2*[F188R], mice with overlapping sequencing peaks at the intended mutation sites were bred with C57BL/6 mice for passing the desired mutations through the germline. For *Sun1*[−/−], heterozygous mouse containing a frameshift at His149 were bred with C57BL/6 mice for passing the desired mutations and homozygous *Sun1*[−/−] mice were confirmed by western blotting.

**Co-IP and western blot.** cDNA fragments were constructed into pEGFP-N1 (Clontech) and transfected into HEK293T cells. After 36 h, cells were collected and lysed in lysis buffer (25 mM Tris-HCl, pH 7.4, 150 mM NaCl, 10% Glycerol, 0.5% TritonX-100, 1 mM EDTA) supplemented with EDTA-free complete protease inhibitor (Roche). Cell lysates were centrifuged and supernatants were precleared and immunoprecipitated with anti-FLAG M2 Affinity Gel (Sigma) at 4 °C with rocking for overnight. Precipitates were then washed with lysis buffer and subjected to sodium dodecyl sulfate-polyacrylamide gel electrophoresis (SDS-PAGE) separation. To detect WT and mutant TERB2 proteins in spermatocytes, testis samples from mouse littermates were homogenized in phosphate-buffered saline (PBS) buffer. Cells were pelleted and lysed by sonication in lysis buffer (25 mM Tris-HCl, pH 7.4, 150 mM NaCl, 10% Glycerol, 0.5% TritonX-100, 1 mM EDTA). Cell lysates were centrifuged and the supernatants were subjected to SDS-PAGE separation. After SDS-PAGE, proteins were blotted onto PVDF membranes (Millipore). The blots were incubated in blocking buffer (5% fat-free milk in PBS buffer supplemented with 0.05% TWEEN-20) at room temperature (RT) for 1 h and incubated with primary antibodies in blocking buffer at 4 °C for overnight. Blots were then washed and incubated in the horseradish peroxidase (HRP)-labeled secondary antibodies at RT for 1 h. After being washed, blots were developed with ECL Prime Western Blotting System (GE Healthcare, RPN2232).

**Hoechst 33342 and propidium iodide (PI) staining for flow cytometry.** Testis cells were prepared for FACS analysis based on previous studies with minor modifications[38,39]. Briefly, decapsulated testes were incubated in DMEM medium supplemented with 1 mg/mL Collagenase type IV (Sigma) and 10 μg/mL DNase I (Roche) at 33 °C for 20 min. Then 1 mg/mL trypsin (Gibco) was added and samples were incubated for another 20 min, with the tube being inverted several times. The

dissociated testis samples were pipetted with a plastic disposable Pasteur pipet several times, followed by adding 10% FBS to inactivate trypsin. For PI staining of fixed cells, testis cell samples were centrifuged and pellets were resuspended in 100 μL PBS and then 900 μL 70% ethanol were added. Cells were fixed overnight at 4 °C, washed once with PBS, resuspended in PI staining solution (PBS supplemented with 0.1% TritonX-100, 200 μg/mL DNase-free RNase and 20 μg/mL PI), incubated at 37 °C for 30 min, and then used for FACS analysis with a BD FACS Calibur Flow Cytometer. For Hoechst 33342 staining, dissociated testis cell samples were incubated with 100 μg/mL Hoechst 33342 and 10 μg/mL DNase I at 33 °C for 1 h and then passed through 40-μm cell strainers (BD Falcon). Cell samples were added with 10 μg/mL PI and immediately subjected to FACS analysis with a BD Influx cell sorter. Examples of the FACS gating strategy are shown in Supplementary Figure 8a, b.

**Antibody production.** The following peptides were chemically synthesized and used for immunization of rabbit: TERB2 (109–125): DEKFQIEKHDRMASSDK, MAJIN (163–186): METSSEASSNKKPLKESKRSRDEE, Speedy A (294–310): NFVKKNKSVEWFAESEE, and SUN1 (82–100): TPAKGRETRTVKQRRSASK. The serums obtained were affinity purified against the corresponding antigens coupled to a HiTrap NHS-activated HP column.

**Antibodies.** The following antibodies were used: rabbit polyclonal antibody against TERB1 (1:1000)[21], TERB2 (this study, 1:2000), MAJIN (this study, 1:2000), SUN1 (this study, 1:1000), Speedy A (this study, 1:3000), phos-HistoneH2A.X (Ser139) (Millipore, 05-636-25UG, 1:2000), SYCP1 (Abcam, ab15087, 1:1000), SYCP3 (Abcam, ab15093, 1:2000), SOX9 (Millipore, ab5535, 1:200), MSH4 (Abcam, ab58666, 1:500), c-Myc (Santa Cruz, sc-789, 1:1000), TRF1 (1:1000)[40], mouse monoclonal antibodies against MLH1 (BD, 550838, 1:200), SYCP3 (Abcam, ab97672, 1:2000), TRF1 (Abcam, ab66223, 1:1000), c-myc (Santa Cruz, sc-40, 1:1000), FLAG (Sigma, F3165, 1:200), actin (Sigma, A2228, 1:2000), DMC1 (Abcam, ab11054, 1:100), Lamin B1 (Proteintech, 66095-1-Ig, 1:500), CDK2 (Santa Cruz, sc-6248, 1:100). Secondary antibody for western blot: Goat anti-Mouse IgG/HRP (Abclonal, AS003, 1:2000), Goat anti-Rabbit IgG/HRP (Proteintech, SA00001, 1:2000). Goat anti-mouse/rabbit secondary antibody for IF: DyLight 488 (Thermo, 35502, 35553, 1:500), DyLight 550 (Thermo, 84540, 84541, 1:500), DyLight 633 (thermos, 35512, 35562, 1:500).

**Quantitative real-time PCR.** Total RNA was purified from seminiferous tubules using Trizol reagent (Invitrogen, 15596026) and cDNA was prepared from RNA using the PrimeScript II 1st Stand cDNA Synthesis Kit (Takara, 6210A). Quantitative real-time PCR was performed with the Power SYBR Green PCR Master Mix (ABI, 4367569) using the ViiA 7 System (Thermo Fisher Scientific). The primers used were as follows: mTerb1-forward: 5′-CAT TCA GTT TGT AAA CAA AAC AGT GAT GCA GGT ATT TA-3′; mTerb1-reverse: 5′-CAA AGA TTG CTG ACA GTA AAC ATT TTC TTC TGC A-3′; mTerb2-forward: 5′-GTG ATA AGG AGA ACA TTA GAC CAA CAC CAG AAC ACA-3′; mTerb2-reverse: 5′-CAA TTG ATA TAT AGC CTG TCA CCA TGT TGT TCA CTG GA-3′; mMajin-forward: 5′-GGC AAC AGT ATC AGA GGA GAA GAA ATA GAG GAT AAG G -3′; mMajin-reverse: 5′-CGG AAG GCA AGT CTT CAG CAA ACC A-3′; mGAPDH-forward: 5′-GGT GAA GGT CGG TGT GAA CG-3′; mGAPDH-reverse: 5′-CTC GCT CCT GGA AGA TGG TG-3′.

**Histological analysis.** The testes and ovaries were dissected and fixed in 4% paraformaldehyde (PFA) in PBS at 4 °C for 8 h. Tissues were dehydrated by treatment with an ethanol series (50%, 75%, 95%, 100%), cleared twice in xylene, and embedded in paraffin. Tissue sections (5 μm) were prepared with a Leica RM2235 rotary microtome. Sections were stained with hematoxylin and eosin according to the standard protocol (BBI, E607318). TUNEL assay was carried out with the TUNEL ApoGreen Detection Kit (Yeasen, 40307ES20).

**Squash and spread preparations.** Squashed and surface-spread spermatocytes were prepared as described previously with minor modifications[41,42]. Briefly, for squashed spermatocytes, freshly extracted 14-dpp (days past partum) testes were minced into small pieces with tweezers in 2% PFA and 0.05% Triton X-100 and

fixed for 10 min. Several small pieces of tubules were then put on a slide previously rinsed with ethanol and chloroform mixture (1:1), covered with a cover slip, and gently pressed to distribute the material. The slide was immersed into liquid nitrogen, and the cover slip was gently removed. The slide was washed in PBS and used immediately for immunostaining. For spermatocyte chromosome spreads, testes were dissected and seminiferous tubules were separated and placed in a hypotonic extraction buffer (30 mM Tris-HCl, pH 8.2, 50 mM sucrose, 17 mM sodium citrate, 5 mM EDTA, 0.5 mM dithiothreitol, and protease inhibitor cocktail (Roche)) for 30 min. Tubules were then moved to 100 mM sucrose solution (pH 8.2) and spermatocytes were detached by pipetting. Drops of cell suspension were then placed on the upper right corner of cover slips (Thermo, 12-545-84) soaked in a 1% PFA solution (Sigma, P6148), pH 9.2, supplemented with 0.15% Triton X-100. Nuclei were dried overnight at RT and ready for immunostaining. Chromosome spreads of oocytes were prepared with 19-dpc (days post coitum) ovaries in the same way as spermatocyte spreads.

**Electron microscopy**. Testes were dissected and incubated in a fixative consisting of 2.5% glutaraldehyde in PBS for 2 h at 4 °C. Samples were washed three times with PBS and fixed in 1% $OsO_4$ in PBS for 2 h at 4 °C and then washed three times with PBS and two times with $ddH_2O$. Pre-embedding staining was performed overnight at 4 °C in the dark using 0.5% uranyl acetate. Samples were washed with $ddH_2O$ and then dehydrated in an ethanol series (25%, 50%, 75%, 95%, and 100%). For plastic embedding, samples were incubated in 3:1 ethanol: resin, 1:1 ethanol: resin, and 1:3 ethanol: resin for 40 min for each, then kept overnight in 100% resin. Samples were put into fresh resin and further permeated for 3 h at RT. Resin polymerization was done at 60 °C for at least 48 h. Resin blocks were cut with an ultra-microtome (Leica) and serial sections of 100-nm thickness each section were collected on formvar copper grid (KYKY) and visualized under an electron microscope (FEI, Talos L120C) with a voltage acceleration of 120 kV.

**IF staining and FISH**. For IF staining of cultured cells, cells were grown on cover slips (Thermo, T_7011254584) and fixed by 4% PFA, permeabilized with 0.15% Triton X-100, blocked with 5% bovine serum albumin (BSA) at RT for 1 h, and incubated with primary antibodies at 4 °C for overnight. For IF staining of spermatocyte spreads, spreads were washed with PBS buffer supplemented with 0.15% Triton X-100, blocked with 5% BSA at RT for 1 h, and incubated with primary antibodies at 4 °C for overnight. For IF staining of paraffin-embedded tissue sections, sections on cover slips were dewaxed twice in fresh xylene for 15 min each and rehydrated in 100%, 95%, 75%, 50% ethanol and PBS for 10 min each. Epitope retrieval was carried out by boiling sections in TUF TARGET UNMASKING FLUID (Invitrogen, Z00R.0000) for 20 min. Sections were then washed with PBST (PBS buffer supplemented with 0.5% Triton X-100) and blocked with 5% BSA at RT for 3 h, followed by incubation with primary antibodies at 4 °C for overnight. Cover slips with cells, sections, or spreads were then washed and incubated with fluorescence-conjugated secondary antibodies at RT for 1 h. Cover slips were thoroughly washed in PBST (PBS buffer supplemented with 0.1% Triton X-100), washed once in PBST supplemented with 1 µg/mL 4,6-diamidino-2-phenylindole (DAPI), air-dried, and ready for microscopic imaging.

IF combined with FISH (IF-FISH) was performed as previously described with minor modifications[43]. Briefly, after being incubated with secondary antibodies, cover slips were washed and fixed with 4% PFA for 10 min, dehydrated in 70%, 85%, and 100% ethanol, and air dried. Samples were denatured at 85 °C for 5 min in presence of Cy3-or Cy5-labeled $(CCCTAA)_4$ PNA probes (TelC) (Panagene) or XMP mouse chromosome 5 or 8 painting probes (Metasystems). Hybridization was then carried out at 37 °C for >12 h. Cover slips were washed once with hybridization buffer, twice with 2×SSC buffer, and once with PBS supplemented with 1 µg/mL DAPI. Cover slips were then air-dried and ready for imaging. Microscopy imaging was performed by LSM 710 (Zeiss) using ×63NA/1.40 oil.

## Data availability

The atomic coordinates and structure factors for the TERB1–TERB2 complex and the TERB2–MAJIN complex have been deposited to the Protein Data Bank (PDB) under the accession code PDB 6J07 and 6J08. The source data underlying Figs. 4b, d, f, g; 5b, f, h; 6b, d; 7b, d, f, h; 8b, d, f; 9b, d; and 10b, d and Supplementary Figs. 2e, 2f, 2g, 4b, 4d, 4f, 5b, 5e, 6b, 6d are provided as a Source Data file. Other data are available from the corresponding authors upon reasonable request.

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

## Acknowledgements

We thank L. Wu, D. Yao, and R. Zhang from BL18U1 and BL19U1 beamlines at NCPSS and Shanghai Synchrotron Radiation Facility (SSRF) for help with crystal data collection and processing. We thank S. He and S. Li from Shanghai Institute of Precision Medicine, and Y. Yu and Y. Wang from NCPSS for help with confocal microscopy and FACS. We thank H. Chen, C. Zheng, and H. Feng from NCPSS for help with mouse breeding. This work was supported by grants from the National Natural Science Foundation of China (U1632267, 31330040, and 31525007 to M.L.), the Strategic Priority Research Program of the Chinese Academy of Sciences (XDB08010201 to M.L.), and the Outstanding Academic Leader Program of Science and Technology Commission of Shanghai Municipality (16XD1405000 to M.L. and C.H.).

## Author contributions

Y.W., Y.C., C.H. and M.L. designed the study. Y.W., Y.C., and C.H. carried out biochemical assays. Y.W., L.N. and L.W. crystallized the TERB1–TERB2 and TERB2–MAJIN complex. J.W., Y.W. and L.W. collected X-ray data and determined the crystal structure. Y.C. and J.C. participated in the generation of the knock-in mice. Y.W., Y.C., J.L., H.C. and C.H. performed the experiments to characterize the meiotic phenotype. Y.W., Y.C., C.H. and M.L. analyzed data. Y.W., C.H., J.W. and M.L. drafted the manuscript. All authors reviewed and edited the manuscript.

## Additional information

**Competing interests:** The authors declare no competing interests.

