## [Peer Review File · Nature Communications]

Reviewers' comments:

Reviewer #1 (Remarks to the Author):

During prophase I of meiosis, the homologous chromosomes pair, synapse, recombine, and segregate to separate poles of the first meiotic spindle. All these processes are challenged by topological problems that may result in illegitimate chromosome interactions that must be resolved. Indeed, if the developing gamete fails in DNA repair and homologous chromosome associations will most likely result in phenotypes ranging from sterility to generation of aneuploid offspring with developmental abnormalities. Therefore, works like this focusing in understanding basic mechanisms on chromosome dynamics in meiosis are very important for the advancement of the field.

The most critical events in meiosis, such as double strand break repair and homologous chromosome interactions, coincide with prominent telomere-led chromosome movements or also called rapid prophase movements (RPMs) that are widely conserved in organisms from yeast to mammals. This type of chromosome movements depend on the connection of telomeres to nuclear envelopment and then to the cytoskeleton via protein bridges through the intact nuclear envelope termed the LINC complex. Using cellular and genetic approaches previous works have shown a strong correlation between the interaction of the meiotic telomeres with the nuclear envelope and RPMs. Importantly, they identified the proteins (TRF1, TERB1, TERB2, and MAJIN) and proposed a mechanism explaining meiotic telomere-nuclear membrane interaction. However, the architecture of this protein complex is not well understood and how they affect in vivo attachment of telomeres, and ultimately how they chromosome dynamics, is poorly understood. Thus, advances in the molecular structure of this complex are necessary to explain functional and regulatory mechanisms for telomere engagement with the nuclear membrane and inter membrane mechanisms directing chromosome movements.

In this work Wang et. al., present: (1) crystal structure of human TERB1-TERB2 and TERB2-MAJIN subcomplexes, (2) Biochemical analysis of TERB1-TERB2 and TERB2-MAJIN interaction, including domain mapping of amino acids responsible of these interactions, (3) and characterization of meiotic phenotype (in vivo studies) of mice carrying mutations that specifically disrupt TERB1-TERB2 and TERB2-MAJIN interactions.

My overall appreciation is that this work will be of interest to a number of researchers in different fields ranging from meiosis to cell nuclear structure and biology. This manuscript contains a substantial amount of good quality work, and most experiments are well focused in supporting a role for the proposed protein interactions in mouse meiosis. However, as presented the manuscript and figures need substantial work before being considered for publication (detailed below).

After the recommended changes, I think this work will be well suited for a high impact journal such as Nature Communications. I have comments and advice for improvement in the written and experimental sections (not indicated in any particular order):

- 1- The following data is important and should be included in the main figures: SF1 on protein interaction and truncations (supporting Fig 1a), SF6a-6c, and SF8a and 8b.
- 2- In several parts of the manuscript data is indicated as "data now shown". All data should be show at least as part of the supplementary figures. I think this is important to support the conclusions and guarantee reproducibility.
- 3- Quantification of some visual data is missing. This is important to support the conclusions, please do: Fig 4a, 4c, SF5a, SF5c, SF6a-6c, 5d, 7a, 5c/SF7b, SF8a and 8b, 6a, SF9a, c, and d.
- 4- I find the current way (structure of the manuscript) in which results are presented confusing. Some restructuring may help understand the result and then conclusions, thus increasing impact. For example, in lines 224-227 they start mentioning defects in chromosome association and even they conclude about defective synapsis. However, this is again described (now in a more proper

way) under the titles in line 315 and 342 (i.e. Fig 7). Data on chromosome synapsis should be unified, come after the description of infertility, and followed by assessment of DNA repair.

5- Data under the title homologous pairing is actually "DSB repair or progression of homologous recombination", and data under title in line 342 is actually "homologous pairing/synapsis". In the way this study analyze homologous chromosomes interactions doesn't really assess differences in both pairing and synapsis.

6- If in lines 315-316 homologous pairing is referring as to pairing of two homologous DNA sequences, this can be easily confused with the concept of homologous chromosome pairing and need to be changed (DSB repair for example).

7- Lines 318-319, the studies here are not necessary revealing synaptic defect consequences. DNA repair defects may be in fact direct consequence of defects in RPMs. It is more adequate to refer to studies in this section as analysis of mutations effect on repair of DSBs thought/using homologous recombination.

8- The MLH1 phenotype in mutants may be caused by early cell arrest at stage of pachytene in which MLH1 has not yet loaded in chromosomal recombination sites. They must assess presence of late pachytene (H1t co-staining) or enforce cell progression using okadaic acid. Then, assess presence of MLH1 on this material. Also, MSH4 immunostaining and quantification of foci per cell should be included in the analysis. MSH4 is an important intermediate marker that may more accurate indicate the defective stage in mutants.

9- Fig 4. The quality of figure 4b should be improved. The image is too small and the contrast is poor to appreciate the conclusion stated in lines 205-208. Also, in order to accurately demonstrate presence of Sertoli and spermatogonia cells, testis slides should be immunostained with specific markers such as Sox9 for Sertoli and Plzf or Stra8 for spermatogonia cells. 4a should include measurement of testis size. In 4c, what types of cells are apoptotic, spermatocytes? If so what stage?

10- Related to statement in line 402. RPMs have not been measured. It would be advisable to measure rapid prophase movements in mutant spermatocytes and compare with wild type cells. RPMs are the ultimate biological consequence of the mutations and should be severely affected by the mutations.

10- In line 276 (in conclusions, and in the abstract). It seems that data is insufficient to support the conclusion that telomere-NE attachment "precedes" stable telomere-LINC. It seems that actually support the idea that they are separate events.

11- How results in line 304-309 are interpreted in the context of the bouquet and prophase cell arrest observed?

11- I do not understand what "the movement of LINC complex at the NE ...independent of the telomere-NE attachment" means. Please, clarify.

12- Lines 108-109. Data actually indicate a model for the structure of the TRF1-TERB1-TERB2-MAJIN complex, but not about how it interacts with the end of chromosomes (i.e. SC or cohesin components) or the LINC. This needs to be toned down.

13- Introduction. Line 53 around the concept of rapid prophase movements and how they contribute to homologous chromosome interactions. The Citations seems inappropriate. There is a number of primary works that may be more appropriate than the cited review. For example, work describing rapid chromosome movements in mouse (Lee et al Cell reports 2015) and, among those few actually showing the consequences of mutations interfering movements, (Lee et al Plos Genetics 2012).

14 - In several parts of the manuscript, i.e. line 76. "atomic picture". It may be more adequate to say "provide support for a molecular model".

15- line 148, symbol is missing.

16- I find the conclusion section not well organized. Perhaps including titles and clearly separating topics will help.

17- Line 370-371. The authors should be more careful (caveat more) respect to what actually are proved functions of chromosome movements. For example, that in mammals RPMs favor chromosome pairing or resolve chromosome entanglements, is only suggested by the current data.

18- Lines 372-374. I suggest: "help reveal at molecular level ..."

19- Line 375- Specifically instead specially.

20- Line 382-387. I am not sure what they mean here. TERB1, TERB2, and MAJIN do not require SUN1.

21- Line 391. "triggers the homologues searching" of Chromosomes? DNA sequences? The phrase is incomplete.

22- Line 393 "homologous searching indicated by the loading of DMC1" Dmc1 loading at chromosomes (foci) in not indication of homologous DNA template searching, particularly given that the authors see similar number of DMC1 foci even homologous chromosome pairing is severely affected by the mutations.

23- Line 394-397. This paragraph is confusing and inaccurate. "chromosome are separated" means they are "unsynapsed"?

24- Clarify STED.

25- Line 410-413. It is not clear what the authors mean with this statement.

26- Line 415-422. This is confusing. Do the authors mean that this is a possible regulatory factor for telomere attachment?

Reviewer #2 (Remarks to the Author):

This work determines using crystallography the structural interfaces between TERB1-TERB2 and TERB2-MAJIN with minimal constructs sufficient to reconstitute the interactions. The interfaces are validated biochemically with co-immunoprecipitation and a large array of in vivo experiments using mice knocked in with interface mutants. In addition to validating the importance of the interfaces in telomere nuclear membrane linkage, the in vivo experiments also suggest that initial membrane attachment of telomeres is independent of the SUN1 protein, and that Speedy A appears to localize with telomeres and nuclear membrane in the presence and absence of proper telomere-nuclear membrane attachment. Overall the in vivo data are designed, executed, and presented with great clarity. It is commendable that the authors validated their crystal structures by generating mouse knock in mutations using CRISPR/Cas9. On the contrary the structural analysis is weak and there are errors in the presentation of biochemical data.

Major points:

1. The crystallographic analysis as presented is not ready for publication. Inspection of the PDB validation report suggests that the quality of the TERB2-MAJIN structure is borderline, while the TERB1-TERB2 structure is even weaker. It doesn't seem as if the low resolution can explain the poor statistics as the structures (especially the TERB1-TERB2) perform poorly even in the context of structures at the same resolution. The authors do show maps in the supplementary data but there seems to be a large disconnect between that and the poor statistics (R_{free} is 35-36%; 3.7% of residues in disallowed Ramachandran space; >10% side chains outliers for the TERB1-TERB2 structure). The authors must work on obtaining higher quality diffraction data to improve the statistics. Regardless of the basis for the current quality of the structures, it seems premature to publish the structural results as is.

2. There are a lot of discrepancies between the definitions of the domains that went into structure determination. Depending on where one looks (methods, main text, PDB validation report), the definitions of the domain boundaries vary drastically. For example Majin's NTD definition varies from amino acid 2 to 149 all the way to 1 to ~110 depending on where it is described. As both the structures are of minimal fragments and not full-length proteins, it is essential to describe

accurately the domain definitions.

3. From the paper describing the discovery of Majin it can be inferred that TERB1-TERB2 is dimeric (2x TERB1 and 2x TERB2). How do the authors reconcile their result that TERB1-TERB2 is a heterodimer (1x TERB1 and 1x TERB2) in the context of what is reported about these proteins in the literature? Is a larger complex of TERB1-TERB2 (as defined in the Majin discovery paper) also a monomer when the authors analyze it by size-exclusion? Or did the process of designing the minimal TERB1-TERB2 constructs result in loss of homodimerization of TERB1-TERB2? Size-exclusion analysis could be used to address this issue.

4. The speculation about Speedy A being a bridge between telomeres and the nuclear membrane is interesting although it has been shown previously that the bridge can occur through a TERB1-SUN1 interaction. How do the authors envision the role of Speedy A in the context of what is reported about TERB1-SUN1?

5. There are issues with the co-IP experiments in Figure 3 b, d, f, and g. All/many of them probably have the Flag and Myc tags on the wrong proteins. Panel d probably has input and beads panels switched; otherwise it is not possible to explain how there can be Flag and Myc signal in all the input lanes in panel d.

Minor points:

1. Fig. 7d is not cited in the text, and neither is the model described. It should either be removed or cited and explained.
2. Line 100 – nulcear should be changed to nuclear.

Point-to-point responses to reviewers' comments

The meiotic TERB1-TERB2-MAJIN complex tethers telomeres to the nuclear envelope NCOMMS-18-27619-T

Reviewer 1

1- The following data is important and should be included in the main figures: SF1 on protein interaction and truncations (supporting Fig 1a), SF6a-6c, and SF8a and 8b.

We have moved data in original SF1 to the revised Fig. 1, and data in original SF8a and SF8b to revised Fig. 9a and Fig 9c, respectively, in the revised manuscript. Due to the limitation of maximal display items number (10 items for *Nat Commu*), data in original SF6a-6c is in revised supplementary Fig. 4 (4a, 4c, and 4e).

2- In several parts of the manuscript data is indicated as “data now shown”. All data should be show at least as part of the supplementary figures. I think this is important to support the conclusions and guarantee reproducibility.

Following this reviewer's suggestion, data mentioned as “data not shown” are now all included in the revised manuscript as listed below.

- (1) Page 8, line 10: “Pups were born close to the expected Mendelian ratio (**Supplementary Table 3**), suggesting that the *Terb2* mutations did not impair normal embryonic development.”.
- (2) Page 8, line 12: “RT-PCR and DNA sequencing were performed to confirm that mRNA transcripts of the mutant *Terb2* genes carried the desired mutations in homozygous testes (**Supplementary Fig. 2b and 2d**).”.
- (3) Page 8, line 19: “Both male and female mutant mice developed normally into adults with no obvious somatic defects. However, repeated breeding attempts of homologous *Terb2*^{Y56E} and *Terb2*^{F188R} male mice with WT female animals did not yield any offspring (**Supplementary Table 4**).”.
- (4) Page 10, line 9: “Like their male siblings, homologous *Terb2*^{Y56E} and *Terb2*^{F188R} female mice were also infertile (**Supplementary Table 4**).”.

3- Quantification of some visual data is missing. This is important to support the conclusions, please

do: Fig 4a, 4c, SF5a, SF5c, SF6a-6c, 5d, 7a, 5c/SF7b, SF8a and 8b, 6a, SF9a, c, and d.

We have included the quantification data in the revised manuscript as listed below.

Original	Revised	Quantification data
Fig4a	Fig. 5a	Fig. 5b
Fig4c	Fig. 5e	Fig. 5f
SF5a	Supplementary Fig.5a	Supplementary Fig. 5b
SF5c	Replaced by Supplementary Fig.5d	Supplementary Fig. 5e
SF6a	Supplementary Fig. 4a	Supplementary Fig. 4b
SF6b	Supplementary Fig. 4c	Supplementary Fig. 4d
SF6c	Supplementary Fig. 4e	Supplementary Fig. 4f
Fig5d	Fig. 8e	Fig. 8f
Fig7a	Fig. 6c	Fig. 6d
Fig5c/SF7b	Fig. 8c	Fig. 8d
SF8a	Fig. 9a	Fig. 9b
SF8b	Fig. 9c	Fig. 9d
Fig6a	Fig. 10a	Fig.10b
Fig6b	Fig. 10c	Fig.10d
SF9a	Fig. 7a	Fig. 7b
SF9c	Fig. 7c	Fig. 7d
SF9d	Fig. 7g	Fig. 7h

4- I find the current way (structure of the manuscript) in which results are presented confusing. Some restructuring may help understand the result and then conclusions, thus increasing impact. For example, in lines 224-227 they start mentioning defects in chromosome association and even they conclude about defective synapsis. However, this is again described (now in a more proper way) under the titles in line 315 and 342 (i.e. Fig 7). Data on chromosome synapsis should be unified, come after the description of infertility, and followed by assessment of DNA repair.

Thanks for this good point. Following this reviewer's suggestion, we have revised the text and rearranged the order of text and figures in the revised manuscript. The result sections are organized as the following.

- (1) Characterization of the TERB1-TERB2 and the TERB2-MAJIN interactions
- (2) Crystal structures of the TERB1-TERB2 and the TERB2-MAJIN complexes
- (3) Mutational analysis of the TERB1-TERB2 and the TERB2-MAJIN interfaces
- (4) Targeted disruption of the TERB1-TERB2 and the TERB2-MAJIN interactions in mice
- (5) The TERB1-TERB2 and the TERB2-MAJIN interactions are required for fertility
- (6) Disruption of the TERB1-TERB2 and the TERB2-MAJIN interactions induces disordered synapsis
- (7) Disruption of the TERB1-TERB2 and the TERB2-MAJIN interactions impairs meiotic DSB repair and homologous recombination
- (8) Disruption of the TERB1-TERB2 and the TERB2-MAJIN interactions abolishes the telomere-NE attachment
- (9) Disruption of the TERB1-TERB2 and the TERB2-MAJIN interactions leads to the congregation of Speedy A and SUN1 at the NE

5- Data under the title homologous pairing is actually "DSB repair or progression of homologous recombination", and data under title in line 342 is actually "homologous pairing/synapsis". In the way this study analyze homologous chromosomes interactions doesn't really assess differences in both pairing and synapsis.

Thanks for this point. We have revised the title (original line 315-316) "Disruption of the TERB1-TERB2 and the TERB2-MAJIN interactions results in aberrant homologous pairing" into "Disruption of the TERB1-TERB2 and the TERB2-MAJIN interactions impairs meiotic DSB repair and homologous recombination." (Page 12, line 4).

6- If in lines 315-316 homologous pairing is referring as to pairing of two homologous DNA sequences, this can be easily confused with the concept of homologous chromosome pairing and need to be changed (DSB repair for example).

We have revised the title (original line 315-316) as "Disruption of the TERB1-TERB2 and the TERB2-MAJIN interactions impairs meiotic DSB repair and homologous recombination." (Page 12, line 4).

7- Lines 318-319, the studies here are not necessary revealing synaptic defect consequences. DNA repair defects may be in fact direct consequence of defects in RPMs. It is more adequate to refer to studies in this section as analysis of mutations effect on repair of DSBs thought/using homologous recombination.

Following this reviewer's suggestion, we have modified the text as "To further investigate the synapsis defects observed in *Terb2*^{Y56E} and *Terb2*^{F188R} male mice, we examined the effects of *Terb2* mutations on homologous recombination-directed DSB repair." (Page 12, line 7).

8- The MLH1 phenotype in mutants may be caused by early cell arrest at stage of pachytene in which MLH1 has not yet loaded in chromosomal recombination sites. They must assess presence of late pachytene (H1t co-staining) or enforce cell progression using okadaic acid. Then, assess presence of MLH1 on this material. Also, MSH4 immunostaining and quantification of foci per cell should be included in the analysis. MSH4 is an important intermediate marker that may more accurately indicate the defective stage in mutants.

Thanks for this good point. We agree that MSH4 is an important intermediate marker to accurately indicate the defective stage in mutant testes. Following this reviewer's suggestion, we have performed IF experiment for MSH4 (revised Fig. 7e and 7f) and revised the paper as the following.

"Next, we analyzed the location of DMC1, a protein involved in early repair of DSBs. In WT zygotene spermatocytes, DMC1 assembles on the axial elements of chromosomes and gradually disappears towards pachynema (Fig. 7c and 7d). However, the *Terb2*^{Y56E} and *Terb2*^{F188R} spermatocytes exhibited persistent DMC1 foci (Fig. 7c and 7d), suggesting that the loading of DMC1 was not affected but subsequent process was arrested in mutant spermatocytes. To further examine the defect of DSB repair in mutant spermatocytes, we analyzed the distributions of MSH4, a marker of intermediate steps in recombination that forms discrete foci along aligned chromosomes, and of MLH1, a key player in the last step of recombination leading to the formation of crossovers. We found that majority of SYCP3 filaments lacked MSH4 foci in the mutant pachytene-like spermatocytes, suggestive of a great reduction of DNA-DNA interaction between homologous chromosomes (Fig 7e and 7f). This is consistent with the result of chromosome painting assays that most chromosomes are spatially separated from their homologs (Supplementary Fig. 6a-d). IF staining revealed no MLH1 foci in the mutant spermatocytes (Fig. 7g and 7h), suggesting a complete absence of Holliday junction formation between homologous chromosomes. Collectively, our data demonstrated that disruption of the TERB1-TERB2-MAJIN complex causes severe impairment of homologous recombination and persistent DSBs." (Page 12, line 16).

9- Fig 4. The quality of figure 4b should be improved. The image is too small and the contrast too poor to appreciate the conclusion stated in lines 205-208. Also, in order to accurately demonstrate presence of Sertoli and spermatogonia cells, testis slides should be immunostained with specific markers such as Sox9 for Sertoli and Plzf or Stra8 for spermatogonia cells. 4a should include measurement of testis size. In 4c, what types of cells are apoptotic, spermatocytes? If so what stage?

We have enlarged the image to improve the quality in the revised Fig 5c (original figure 4b). We also performed new IF experiment for SOX9 to show Sertoli cells (revised Fig 5d) and included the measurement of testis size (revised Fig 5b).

In original Fig. 4c (revised Fig. 5e), the apoptotic cells are spermatocytes at late zygotene-early pachytene stage. This conclusion was based on the following observations. First, the number of Sertoli cells in mutant seminiferous tubule remains roughly the same as in WT seminiferous tubule (revised Fig 5d). Second, leptotene and zygotene spermatocytes are evident by IF staining of SYCP3 and SYCP1 (revised Fig 5g). Third, we did not observe typical pachytene spermatocytes that display completely aligned chromosomes (revised Fig 5g and Fig 7a).

10- Related to statement in line 402. RPMs have not been measured. It would be advisable to measure rapid prophase movements in mutant spermatocytes and compare with wild type cells. RPMs are the ultimate biological consequence of the mutations and should be severely affected by the mutations.

In this work, we revealed the molecular basis of the interactions among the TERB1-TERB2-MAJIN complex, and demonstrated that point mutations of TERB2 that disrupt either the TERB1-TERB2 or the TERB2-MAJIN interaction completely abolished the connection between telomeres and the NE. In this regard, these point mutations have exactly the same effect as TERB2 knockout in disrupting the telomere-NE connection as shown in the Shibuya paper (Shibuya, Hernandez-Hernandez et al. 2015). In the 2015 Cell paper, Shibuya et al clearly showed that knockout of TERB2 abolishes chromosome rapid prophase movements (Shibuya, Hernandez-Hernandez et al. 2015). Based on these observations, it is easy to conclude that RPMs should also be severely affected by the TERB2 point mutations. Therefore, we think that it is not necessary to repeat the RPM experiment.

11- In line 276 (in conclusions, and in the abstract). It seems that data is insufficient to support the conclusion that telomere-NE attachment “precedes” stable telomere-LINC. It seems that actually support the idea that they are separate events.

Following this reviewer’s suggestion, we have modified the text at the end of the “Disruption of the TERB1-TERB2 and the TERB2-MAJIN interactions abolishes the telomere-NE attachment” section as “Together, we conclude that both the TERB1-TERB2-MAJIN complex and the LINC complex contribute to the stable telomere-NE association.” (Page 14, line 13).

In the end of the “Both the TERB1-TERB2-MAJIN complex and the LINC complex are required for the stable telomere-NE connection” section in Discussion, we emphasize this point as “Taken together, we propose that the telomere-TRF1-TERB1-TERB2-MAJIN-NE interaction network and the telomere-LINC complex connection are likely two separate but cooperative pathways to stably recruit telomeres to the NE in meiosis prophase I...” (Page 17, line 8).

We also revised the manuscript as “In addition, depletion of SUN1 also partially disrupts the telomere-NE connection. we propose that the telomere-TRF1-TERB1-TERB2-MAJIN-NE interaction network and the telomere-LINC complex connection are likely two separate but cooperative pathways to stably recruit telomeres to the NE in meiosis prophase I.” (Page 1, line 7).

12- How results in line 304-309 are interpreted in the context of the bouquet and prophase cell arrest observed?

In WT spermatocytes, telomeres cluster together to form the bouquet during a short time window in late zygotene stage, and disperse along the nuclear envelope after early pachytene. Because telomere bouquet formation is a transient event, only a small population of nuclei (~4%) exhibits congregation of SUN1, Speedy A and telomeres (revised Fig. 10a-d). In *Terb2* mutant spermatocytes, the presence of SUN1 congregation indicates that SUN1 (presumably the LINC complex) moving and clustering on the NE in early prophase I despite of the absence of telomere-NE connection (revised Fig. 10c). However, the population of nuclei displaying congregation of SUN1 in mutant mice is remarkably higher (~35%) compared to that in WT ones, suggesting that SUN1 dispersion is severely delayed or arrested in mutant nuclei (revised Fig. 10c and 10d). It is likely that in *Terb2* mutant spermatocytes persistent DSBs and massive disordered chromosome synapsis may lead to a checkpoint to prevent the dispersion of the LINC complex and initiate an arrest of prophase progression.

13- I do not understand what “the movement of LINC complex at the NE ...independent of the telomere-NE attachment” means. Please, clarify.

In this study we showed that TERB2 mutations that disrupt the TERB1-TERB2 and TERB2-MAJIN interactions completely abolish the telomere-NE connection (revised Fig. 8a and 8b). Furthermore, we also showed that in *Terb2* mutant spermatocytes, the presence of SUN1 congregation is similar to what is observed in WT spermatocytes (revised Fig. 10c). Therefore, we conclude that SUN1 moving and clustering on the NE in early prophase I does not require the connection between telomeres and the NE.

We have revised the text as “Notably, congregation of SUN1 at the NE was also observed in both *Terb2*^{Y56E} and *Terb2*^{F188R} nuclei, suggesting that SUN1 moving and clustering on the NE in early prophase I is independent of the telomere-NE connection (revised Fig. 10c).” (Page 15 bottom and

Page 16 top)

14- Lines 108-109. Data actually indicate a model for the structure of the TRF1-TERB1-TERB2-MAJIN complex, but not about how it interacts with the end of chromosomes (i.e. SC or cohesin components) or the LINC. This needs to be toned down.

We revised the sentence as “Collectively, these results reveal a linear interaction network within the TERB1-TERB2-MAJIN complex (Fig. 2a).” in the revised manuscript. (Page 4, bottom).

15- Introduction. Line 53 around the concept of rapid prophase movements and how they contribute to homologous chromosome interactions. The citations seems inappropriate. There is a number of primary works that may be more appropriate than the cited review. For example, work describing rapid chromosome movements in mouse (Lee et al Cell reports 2015) and, among those few actually showing the consequences of mutations interfering movements, (Lee et al Plos Genetics 2012).

We have changed the citations as suggested in the revised manuscript.

16- In several parts of the manuscript, i.e. line 76. “atomic picture”. It may be more adequate to say “provide support for a molecular model”.

Following this reviewer’s suggestion, we revised the sentences as follows:

(Page 1, line 10), “...**provides a molecular model** of the connection between telomeres and the NE and reveals the correlation between aberrant synapsis and the defective telomere attachment to the NE.”

(Page 3, line 18), “...**provides a molecular model** of the connection between telomeres and the NE via the TRF1-TERB1-TERB2-MAJIN interaction pathway and reveals the correlation between aberrant synapsis and the defective telomere attachment to the NE.”

(Page 6, bottom), “...**provide a molecular model** of the telomere-NE connection through the TRF1-TERB1-TERB2-MAJIN interaction network (Fig. 3e).”

17- line 148, symbol is missing.

Corrected.

18- I find the conclusion section not well organized. Perhaps including titles and clearly separating topics will help.

We have revised the discussion section as suggested.

19- Line 370-371. The authors should be more careful (caveat more) respect to what actually are proved functions of chromosome movements. For example, that in mammals RPMs favor chromosome pairing or resolve chromosome entanglements, is only suggested by the current data.

We have revised the text as “During meiotic prophase I, telomeres attach to and move on the NE, leading chromosome movements, a dynamic process essential for meiosis.” (Page 16, line 11)..

20- Lines 372-374. I suggest: “help reveal at molecular level ...”

Following this reviewer’s suggestion, we have revised the sentence as “In this study, we determined the crystal structures of the human TERB1_{T2BM}-TERB2_{NTD} and TERB2_{MBM}-MAJIN_{NTD} complexes, and **helped reveal at molecular level** how these proteins establish the telomere-NE connection.” (Page 16, line 14).

21- Line 375- Specifically instead specially.

Corrected.

22- Line 382-387. I am not sure what they mean here. TERB1, TERB2, and MAJIN do not require SUN1.

Thanks for this point. We have completely rewrote the discussion about the TERB1-TERB2-MAJIN complex and the LINC complex (Pages 16-17). We propose that the telomere-TRF1-TERB1-TERB2-MAJIN-NE interaction network and the telomere-LINC complex connection are likely two separate but cooperative pathways to stably recruit telomeres to the NE in meiosis prophase I.

23- Line 391. “triggers the homologues searching” of Chromosomes? DNA sequences? The phrase is incomplete.

We have modified the text as “During WT early prophase I, the conserved programmed DSB repair triggers a genome wide search for DNA sequence homology.” (Page 18, line 13).

24- Line 393 “homologous searching indicated by the loading of DMC1” Dmc1 loading at chromosomes (foci) in not indication of homologous DNA template searching, particularly given that

the authors see similar number of DMC1 foci even homologous chromosome pairing is severely affected by the mutations.

Thanks for this good point. We revised the manuscript as "...loading of DMC1 on the chromosomes suggests that DNA-RAD51 complexes were formed in mutant spermatocytes (Fig. 7c and 7d). However, several lines of evidence suggest that the subsequent DSB repair process was severely impaired. First, the DMC1 signal is persistently remained on the chromosomes in mutant pachytene-like spermatocytes (Fig. 7c and 7d)." (Page 18, line 18).

25- Line 394-397. This paragraph is confusing and inaccurate. "chromosomes are separated" means they are "unsynapsed"?

FISH chromosome painting is a method to monitor the chromosome territory (Ding, Xu et al. 2007, Ishiguro, Kim et al. 2014). Here we used this method to exam whether homologous chromosomes are associated together in mutant spermatocytes (revised SF6a-6d).

In Page 11 line 4, we revised the manuscript as "To investigate the underlying mechanism for the synapsis defects in mutant mice, we performed FISH chromosome painting to examine the homologous association in spermatocytes. Majority of the arrested mutant spermatocytes exhibited separated chromosome signals (Supplementary Fig. 6a-d), consistent with the presence of large number of SYCP3 filaments (Fig. 6a). FISH experiments with probes to chromosomes 5 and 8 revealed similar results (Supplementary Fig. 6a-d), suggesting that defect of homologous association is not specific to particular chromosomes."

26- Clarify STED.

The full name of STED is included in the revised text – "The aberrant synapsis in the mutant spermatocytes was confirmed by the stimulated emission depletion (STED) super-resolution microscopy." (Page 11, line 19).

27- Line 410-413. It is not clear what the authors mean with this statement.

We agree with the reviewer that this sentence is not clear. So we removed it in the revised manuscript.

28- Line 415-422. This is confusing. Do the authors mean that this is a possible regulatory factor for telomere attachment?

Data presented here and in the original papers that identified TERB1, TERB2 and MAJIN (Shibuya,

Hernandez-Hernandez et al. 2015) clearly demonstrate that the TRF1-TERB1-TERB2-MAJIN interaction network is essential for the telomere-NE connection. However, our data and studies from other groups also showed that only this interaction network is not enough for the stable telomere-NE association (revised Fig. 8a and 8b) (Ding, Xu et al. 2007, Viera, Alsheimer et al. 2015, Mikolcevic, Isoda et al. 2016). SUN1 is required for the stable NE attachment of telomeres, as deletion of SUN1 results in substantial increase of internal telomeres at the equatorial region of squashed spermatocytes (revised Fig. 8a and 8b) (Ding, Xu et al. 2007). In the 2014 NCB paper, Shibuya et al detected an interaction between TERB1 and SUN1 through yeast two-hybrid assay (Shibuya, Ishiguro et al. 2014). However, we could not detect any interaction between TERB1 and SUN1 using several different biochemical and biophysical assays including yeast two-hybrid, Co-IP and ITC. Speedy A has previously been shown to localize at telomeres in prophase I (Mikolcevic, Isoda et al. 2016). Moreover, lack of Speedy A severely impairs meiotic telomere-NE attachment (Mikolcevic, Isoda et al. 2016, Tu, Bayazit et al. 2017), suggesting that Speedy A could play an important role in telomere-NE attachment. In the present study, we found two pools of Speedy A—one with telomeres in the nucleoplasm and the other with the LINC complex at the NE—when the telomere-NE connection is disrupted (revised Fig. 9a and Fig. 10a). This result strongly suggests that Speedy A is likely capable of interacting with both telomeres and the LINC complex. A recent study identified a direct interaction between Speedy A and TRF1 (Wang, Tu et al. 2018), supporting this idea.

Reviewer 2

1. The crystallographic analysis as presented is not ready for publication. Inspection of the PDB validation report suggests that the quality of the TERB2-MAJIN structure is borderline, while the TERB1-TERB2 structure is even weaker. It doesn't seem as if the low resolution can explain the poor statistics as the structures (especially the TERB1-TERB2) perform poorly even in the context of structures at the same resolution. The authors do show maps in the supplementary data but there seems to be a large disconnect between that and the poor statistics (Rfree is 35-36%; 3.7% of residues in disallowed Ramachandran space; >10% side chains outliers for the TERB1-TERB2 structure). The authors must work on obtaining higher quality diffraction data to improve the statistics. Regardless of the basis for the current quality of the structures, it seems premature to publish the structural results as is.

Following this reviewer's suggestion, we have tried our best to obtain better datasets using a variety of methods such as optimizing crystallization conditions, using more cryo-protectants, testing crystal dehydration and trying different annealing time, etc. However, these efforts generated no improvement of the crystal diffraction limit. We then carried out careful further refinement of two structures and the

related statistics have been greatly improved as shown below.

1) The R_{work}/R_{free} values were improved from 29.9%/35.1% to 28.9%/33.7% for the TERB1-TERB2 complex and from 23.0%/27.9% to 21.2%/26.2% for the TERB2-MAJIN complex. The TERB1-TERB2 complex was crystalized with a very high solvent fraction (78%), We believe this is the reason for a relatively high R_{work} and R_{free} values. The electron density map is clear enough to allow us to confidently build the model of the TERB1-TERB2 complex (please see the revised Supplementary Figure 1a and 1c).

2) The favored residues in Ramachandran plot were improved from 84.0% to 91.4% (the TERB1-TERB2 complex) and from 90.7% to 93.4% (the TERB2-MAJIN complex), and there were no outlier residues in the structures after refinement.

3) All sidechain outliers had been completely eliminated in both structures.

The statistics mentioned above and all the other related information with the refinement could be tracked in the re-submitted Supplementary Table and Validation Reports.

Based on the refined coordinate and structure factor of the TERB1-TERB2 complex, two stereo-view figures of the density map were generated. One is centered at the zinc-figure region of TERB1 (**Reviewer Figure 1**) and the other at the interface between TERB2 and TERB1 (**Reviewer Figure 2**). Although the resolution of this structure is about 3.3 Å, the quality of the density map was sufficient to trace unambiguously almost all the residues in the TERB1-TERB2 binary complex and to provide structural information for mutation design used in following cell and animal experiments.

Reviewer Figure 1. Stereo view of the electron density map of the zinc-finger region in TERB1 ($\sigma = 1$).

Reviewer Figure 2. Stereo view of the electron density map of the interface between TERB2 and TERB1($\sigma = 1$).

2. There are a lot of discrepancies between the definitions of the domains that went into structure determination. Depending on where one looks (methods, main text, PDB validation report), the definitions of the domain boundaries vary drastically. For example, Majin's NTD definition varies from amino acid 2 to 149 all the way to 1 to ~110 depending on where it is described. As both the structures are of minimal fragments and not full-length proteins, it is essential to describe accurately the domain definitions.

Thanks for pointing out this issue. We mapped the interactions domains by BiFC, NE-localization and gel filtration assays and found that TERB1₅₈₇₋₆₃₆ and TERB2₁₋₁₁₆ mediate the TERB1-TERB2 interaction, and TERB2₁₆₈₋₂₀₂ and MAJIN₁₋₁₄₉ mediate the TERB2-MAJIN interaction (revised Fig 1a-e). To reveal the structural basis for the interactions between TERB1 and TERB2 and between TERB2 and MAJIN, we crystallized the TERB1₅₈₇₋₆₅₉-TERB2₂₋₁₁₆ and the TERB2₁₁₉₋₂₂₀-MAJIN₂₋₁₄₉ complexes and determined their structures. To be sure to include all the interacting regions, protein fragments used in the structural study were larger than those identified based on the mapping assays. Well-defined electron density maps allowed unambiguous tracing of polypeptide chains TERB1₅₉₀₋₅₄₉, TERB2₄₋₁₁₀, TERB2₁₇₄₋₂₀₉ and MAJIN₂₋₁₀₉ in the complexes (revised Supplementary Fig. 1a and 1b), whereas the rest of residues are disordered in the crystals and presumably dispensable for the interactions. Notably, the interaction domains revealed by the crystal structures are consistent with those identified by the mapping analyses (revised Fig. 1a-e). Based on these results, we define TERB2₁₋₁₁₆ as TERB2_{NTD} (N-terminal domain), TERB1₅₈₇₋₆₃₆ as TERB1_{T2BM} (TERB2-binding motif), MAJIN₁₋₁₁₀ as MAJIN_{NTD}, and TERB2₁₆₈₋₂₀₂ as TERB2_{MBM} (MAJIN1-binding motif), respectively (revised Fig. 2a).

The interaction domains are now accurately defined in the revised manuscript.

3. From the paper describing the discovery of Majin it can be inferred that TERB1-TERB2 is dimeric (2x TERB1 and 2x TERB2). How do the authors reconcile their result that TERB1-TERB2 is a heterodimer (1x TERB1 and 1x TERB2) in the context of what is reported about these proteins in the literature? Is a larger complex of TERB1-TERB2 (as defined in the Majin discovery paper) also a monomer when the authors analyze it by size-exclusion? Or did the process of designing the minimal TERB1-TERB2 constructs result in loss of homodimerization of TERB1-TERB2? Size-exclusion analysis could be used to address this issue.

In the 2015 Cell paper, Shibuya et al showed that a C-terminal fragment of mouse mTERB1 (mTRFB1_C, residues 523-699, equivalent to human TERB1 residues 522-661) could form a homodimer in the absence of TRF1 (Shibuya, Hernandez-Hernandez et al. 2015). In our study, we used residues 587-659 of human TERB1 and residues 2-116 of human TERB2 to form the TERB1-TERB2 complex. Static Light Scattering analysis showed that the TERB1₅₈₇₋₆₅₉-TERB2₂₋₁₁₆ complex has a molecular weight corresponding to a 1:1 stoichiometry (**Reviewer Figure 3**). Therefore, it is possible that residues 523-587 could potentially mediate the dimerization of TERB1 (**Reviewer Figure 3**).

Reviewer Figure 3. Biochemical analysis of the TERB1₅₈₇₋₆₅₉-TERB2₂₋₁₁₆ complex.

We have reported the crystal structure of the complex between the TRFH domain of TRF1 and the

TRF1-binding motif of TERB1 (TBM, residues 642-656) (Long, Huang et al. 2017). We showed that the TRF1_{TRFH}-TERB1_{TBM} complex forms a 2:2 heterotetramer. Therefore, full-length TRF1 and TERB1 could form a complex with a 2:2 stoichiometry, and TRF1 and TERB1-TERB2-MAJIN form a super complex with a 2:2:2:2 stoichiometry. This issue is not related to the current study and will be worthy of further investigation in a separate study.

4. The speculation about Speedy A being a bridge between telomeres and the nuclear membrane is interesting although it has been shown previously that the bridge can occur through a TERB1-SUN1 interaction. How do the authors envision the role of Speedy A in the context of what is reported about TERB1-SUN1?

Data presented here and in the original papers that identified TERB1, TERB2 and MAJIN (Shibuya, Hernandez-Hernandez et al. 2015) clearly demonstrate that the TRF1-TERB1-TERB2-MAJIN interaction network is essential for the telomere-NE connection. However, our data and studies from other groups also showed that only this interaction network is not enough for the stable telomere-NE association (revised Fig. 8a and 8b) (Ding, Xu et al. 2007, Viera, Alsheimer et al. 2015, Mikolcevic, Isoda et al. 2016). SUN1 is required for the stable NE attachment of telomeres, as deletion of SUN1 results in substantial increase of internal telomeres at the equatorial region of squashed spermatocytes (revised Fig. 8a and 8b) (Ding, Xu et al. 2007). In the 2014 NCB paper, Shibuya et al detected an interaction between TERB1 and SUN1 through yeast two-hybrid assay (Shibuya, Ishiguro et al. 2014). However, we could not detect any interaction between TERB1 and SUN1 using several different biochemical and biophysical assays including yeast two-hybrid, Co-IP and ITC. Speedy A has previously been shown to localize at telomeres in prophase I (Mikolcevic, Isoda et al. 2016). Moreover, lack of Speedy A severely impairs meiotic telomere-NE attachment (Mikolcevic, Isoda et al. 2016, Tu, Bayazit et al. 2017), suggesting that Speedy A could play an important role in telomere-NE attachment. In the present study, we found two pools of Speedy A—one with telomeres in the nucleoplasm and the other with the LINC complex at the NE—when the telomere-NE connection is disrupted (revised Fig. 9a and Fig. 10a). This result strongly suggests that Speedy A is likely capable of interacting with both telomeres and the LINC complex. A recent study identified a direct interaction between Speedy A and TRF1 (Wang, Tu et al. 2018), supporting this idea.

5. There are issues with the co-IP experiments in Figure 3 b, d, f, and g. All/many of them probably have the Flag and Myc tags on the wrong proteins. Panel d probably has input and beads panels

switched; otherwise it is not possible to explain how there can be Flag and Myc signal in all the input lanes in panel d.

We apologize for the confusion and thank the reviewer very much to point out this error. In the Figure 3b, 3d, 3f and 3g (revised Fig. 4b, 4d, 4f and 4g), we used Flag-TERB2 to Co-IP Myc-TERB2. We have corrected the labeling. In Panel d (original Fig. 3d), we made a mistake when we cropped the image for “anti-Flag” in input lanes. We have corrected this mistake in the revised Fig. 4d. Please see the uncut gel images below for the revised Fig. 4d.

Minor points:

1. Fig. 7d is not cited in the text, and neither is the model described. It should either be removed or cited and explained.

We have removed the original Fig. 7d in the revised manuscript.

2. Line 100 – nulcear should be changed to nuclear.

Thanks for pointing out this typo. We have corrected it in the revised manuscript.

Reference

- Ding, X., R. Xu, J. Yu, T. Xu, Y. Zhuang and M. Han (2007). "SUN1 is required for telomere attachment to nuclear envelope and gametogenesis in mice." *Dev Cell* **12**(6): 863-872.
- Ishiguro, K., J. Kim, H. Shibuya, A. Hernandez-Hernandez, A. Suzuki, T. Fukagawa, G. Shioi, H. Kiyonari, X. C. Li, J. Schimenti, C. Hoog and Y. Watanabe (2014). "Meiosis-specific cohesin mediates homolog recognition in mouse spermatocytes." *Genes Dev* **28**(6): 594-607.
- Long, J., C. Huang, Y. Chen, Y. Zhang, S. Shi, L. Wu, Y. Liu, C. Liu, J. Wu and M. Lei (2017). "Telomeric TERB1-TRF1 interaction is crucial for male meiosis." *Nat Struct Mol Biol* **24**(12): 1073-1080.
- Mikolcevic, P., M. Isoda, H. Shibuya, I. Del Barco Barrantes, A. Igea, J. A. Suja, S. Shackleton, Y. Watanabe and A. R. Nebreda (2016). "Essential role of the Cdk2 activator RingoA in meiotic telomere tethering to the nuclear envelope." *Nat Commun* **7**: 11084.
- Shibuya, H., A. Hernandez-Hernandez, A. Morimoto, L. Negishi, C. Hoog and Y. Watanabe (2015). "MAJIN Links Telomeric DNA to the Nuclear Membrane by Exchanging Telomere Cap." *Cell* **163**(5): 1252-1266.
- Shibuya, H., K. Ishiguro and Y. Watanabe (2014). "The TRF1-binding protein TERB1 promotes chromosome movement and telomere rigidity in meiosis." *Nat Cell Biol* **16**(2): 145-156.
- Tu, Z., M. B. Bayazit, H. Liu, J. Zhang, K. Busayavalasa, S. Risal, J. Shao, A. Satyanarayana, V. Coppola, L. Tessarollo, M. Singh, C. Zheng, C. Han, Z. Chen, P. Kaldis, J. A. Gustafsson and K. Liu (2017). "Speedy A-Cdk2 binding mediates initial telomere-nuclear envelope attachment during meiotic prophase I independent of Cdk2 activation." *Proc Natl Acad Sci U S A* **114**(3): 592-597.
- Viera, A., M. Alsheimer, R. Gomez, I. Berenguer, S. Ortega, C. E. Symonds, D. Santamaria, R. Benavente and J. A. Suja (2015). "CDK2 regulates nuclear envelope protein dynamics and telomere attachment in mouse meiotic prophase." *J Cell Sci* **128**(1): 88-99.
- Wang, L., Z. Tu, C. Liu, H. Liu, P. Kaldis, Z. Chen and W. Li (2018). "Dual roles of TRF1 in tethering telomeres to the nuclear envelope and protecting them from fusion during meiosis." *Cell Death Differ*: doi:10.1038/s41418-41017-40037-41418.

REVIEWERS' COMMENTS:

Reviewer #1 (Remarks to the Author):

The authors have done an exceptional good effort in answering the concerns of this reviewer. They performed experiments and modified the text extensively. In my opinion, the paper is ready for publication.

A minor observation: Point 8. 5th line from bottom. "IF staining...." Should be: As expected, IF staining....

Also, replace Holliday junction for crossovers.

Reviewer #2 (Remarks to the Author):

The major issues raised in the first review including the low quality of X-ray data and issues with the coimmunoprecipitation experiments have been addressed. Although the R-factors are still on the high side, the extensive in vivo validation of the interfaces suggest that the structures are of sufficient quality to reach the conclusions of the manuscript. The authors' willingness to share the PDBs before publication is much appreciated. It seems in the TERB2-MAJIN asymmetric unit there are three protomers. Protomers B and C have quite an extensive interface. Protomer A in two adjacent asymmetric units also seem to form the same kind of interface. While it may be beyond the scope of this study to validate the potential dimeric interface, the authors should at least in the main figure (and in supplement if necessary) show the asymmetric unit and point to the possible dimeric complex of MAJIN-TERB2.

Response to reviewers' comments

Reviewer #1

The authors have done an exceptional good effort in answering the concerns of this reviewer. They performed experiments and modified the text extensively. In my opinion, the paper is ready for publication.

Thanks!

A minor observation: Point 8. 5th line from bottom. "IF staining...." Should be: As expected, IF staining....

Corrected.

Also, replace Holliday junction for crossovers.

Corrected.

Reviewer #2

The major issues raised in the first review including the low quality of X-ray data and issues with the coimmunoprecipitation experiments have been addressed. Although the R-factors are still on the high side, the extensive *in vivo* validation of the interfaces suggest that the structures are of sufficient quality to reach the conclusions of the manuscript. The authors' willingness to share the PDBs before publication is much appreciated. It seems in the TERB2-MAJIN asymmetric unit there are three protomers. Protomers B and C have quite an extensive interface. Protomer A in two adjacent asymmetric units also seem to form the same kind of interface. While it may be beyond the scope of this study to validate the potential dimeric interface, the authors should at least in the main figure (and in supplement if necessary) show the asymmetric unit and point to the possible dimeric complex of MAJIN-TERB2.

Thanks for this good point. We added the description of the observed TERB2_{MBM}-MAJIN_{NTD} heterotetramer in the revised manuscript as the following (Pages 6-7).

“It is worthy of noticing that, although there are three TERB2_{MBM}-MAJIN_{NTD} complexes in one asymmetric unit in the crystal structure, two protomer complexes apparently form a heterotetramer with an extensive hydrophobic dimeric interface between two MAJIN_{NTD} molecules (Fig. 3e and Supplementary Fig. 1e and 1f). Interestingly, the third protomer form a second heterotetramer with another complex from an adjacent unit cell with the identical hydrophobic interface (Supplementary Fig. 1g). Consistent with a recent study, this observation suggests that TERB2 and MAJIN could form a heterotetramer with a 2:2 stoichiometry in cells. Notably, TRF1 itself forms a stable dimer and binds two TERB1 molecules through its TRFH domain in cells. Therefore, TRF1, TERB1, TERB2 and MAJIN can potentially form a large complex with two sets of each protein with multiple dimeric interfaces (Fig. 3f).”